# An ethnographic study of how health system, socio-cultural and individual factors influence uptake of intermittent preventive treatment of malaria in pregnancy with sulfadoxine-pyrimethamine in a Ghanaian context

Matilda Aberese-Ako[1]*, Pascal Magnussen[2], Gifty D. Ampofo[3], Margaret Gyapong[1], Evelyn Ansah[1], Harry Tagbor[3]

1 Institute of Health Research, University of Health and Allied Sciences, Ho, Volta Region, Ghana, 2 Faculty of Health and Medical Sciences, Centre for Medical Parasitology, University of Copenhagen, Copenhagen, Denmark, 3 School of Medicine, University of Health and Allied Sciences, Ho, Volta Region, Ghana

* maberese-ako@uhas.edu.gh, maberese@gmail.com

## Abstract

### Background

Intermittent preventive treatment of malaria among pregnant women with sulfadoxine-pyrimethamine (IPTp-SP), is one of the three recommended interventions for the prevention of malaria in pregnancy (MiP) in sub-Sahara Africa. The World Health Organisation recommended in 2012 that SP be given at each scheduled ANC visit except during the first trimester and can be given a dose every month until the time of delivery, to ensure that a high proportion of women receive at least three doses of SP during pregnancy. Despite implementation of this policy, Ghana did not attain the target of 100% access to IPTp-SP by 2015. Additionally, negative outcomes of malaria infection in pregnancy are still recurring. This ethnographic study explored how health system, individual and socio-cultural factors influence IPTp-SP uptake in two Ghanaian regions.

### Methods

The study design was ethnographic, employing non-participant observation, case studies and in depth interviews in 8 health facilities and 8 communities, from April 2018 to March 2019, in two Ghanaian regions. Recommended ethical procedures were observed.

### Results

Health system factors such as organization of antenatal care (ANC) services and strategies employed by health workers to administer SP contributed to initial uptake. Women's trust in the health care system contributed to continued uptake. Inadequate information provided to women accessing ANC, stock-outs and fees charged for ANC services reduced access to IPTp-SP. Socio-cultural factor such as encouragement from social networks influenced

**Data Availability Statement:** There are legal restrictions on sharing de-identified data. Data

would be provided upon reasonable request from
the ethics committee. Contact information for
ethics committee is as follows: Mr. Fidelis Anumu,
University of Health and Allied Sciences Research
Ethics Committee administrator, email: rec@uhas.
edu.gh.

**Funding:** This work was supported through the
DELTAS Africa Initiative [MARCAD Grant Number:
DEL-15-010]. The DELTAS Africa Initiative is an
independent funding scheme of the African
Academy of Sciences (AAS)'s Alliance for
Accelerating Excellence in Science in Africa (AESA)
and supported by the New Partnership for Africa's
Development Planning and Coordinating Agency
(NEPAD Agency) with funding from the Wellcome
Trust [WT: 107741/A/15/Z] and the UK
government. Disclaimer.

**Competing interests:** The authors have declared
that no competing interests exist.

utilization of ANC services and IPTp-SP uptake. Individual factors such as refusing to take
SP, skipping ANC appointments and initiating ANC attendance late affected uptake.

## Conclusion

Health system, socio-cultural and individual factors influence uptake of optimum doses of
IPTp-SP. Consequently, interventions that aim at addressing IPTp-SP uptake should focus
on regular and sufficient supply of SP to health facilities, effective implementation of free
ANC, provision of appropriate and adequate information to women and community outreach
programmes to encourage early and regular ANC visits.

## Background

The effectiveness of intermittent preventive treatment of malaria among pregnant women
(IPTp), with sulfadoxine-pyrimethamine (IPTp-SP) has been established [1–6]. Since 2012 the
WHO recommended that SP should be given at each scheduled ANC visit except during the
first trimester. Subsequently, SP can be given every month until the time of delivery, with
doses given at least one month apart. This is to ensure that a high proportion of women receive
at least three doses of SP during pregnancy [7, 8]. Consequently, most malaria endemic sub-
Saharan African countries have adopted the World Health Organization's (WHO) recommen-
dation of IPTp-SP as one of the key interventions [1, 9–17].

Several studies have been carried out over the years to evaluate implementation success of
IPTp-SP in the sub region [1, 9, 10, 13–15, 18, 19]. Some of the studies reported that failure to
inform women of additional doses of IPTp-SP and to offer them additional doses on subse-
quent ANC visits contributed to low uptake [19, 20]. Hill, Dellicour's [19] study has contrib-
uted to understanding health worker interaction with ANC clients in the process of offering
IPTp-SP service through reported experiences, thus this current study leverages on their study
to report on observed interactions between health care providers and women in the ANC dur-
ing the provision of IPTp-SP. Also, studies have reported that possible causes of low uptake
could be poor health provider supervision and lack of enforcement of the direct observation
therapy (DOT) in the administration of IPTp-SP [9, 18]. Other challenges that have been
noted are stock-outs of SP, confusion over number of doses that should be provided and poor
dissemination of focused ANC guidelines [10, 13, 21]. Additionally, late booking for ANC,
poor staff attitudes and motivation and IPTp-SP intervention not systematically being
enforced like other interventions have been reported [10, 13, 21].

Ghana instituted a nationwide fee-free maternal healthcare policy intervention in 2004
[22–24] and subsequently integrated the WHO's IPTp-SP intervention [17, 25–27]. Ghana
later revised the policy by extending the minimum number of doses to 5 and maximum 7
doses in 2014 [28]. The country recorded 97% of women attending ANC in 2014 [29] and 98%
in 2017 [30]. Also, the number of women in Ghana who reported that they attended ANC at
least four times increased from 78% in 2008 to 87% in 2014 [29, 31] and to 98% in 2017 [30].
This has made the integrated antenatal care (ANC) service delivery an ideal platform for the
implementation of IPTp-SP in Ghana. So it is expected that ANC attendants should be able to
obtain the minimum 5 doses, as some of the factors that have been reported to determine
adherence to IPTp-SP uptake are the number of ANC visits and knowledge of malaria [13, 32–
34]. Yet, the National Malaria Control Programme [22] reported that in 2017, 68.3% of ANC
registrants received the first dose of SP, compared to 64.0% in 2016 and 57.8% took the second

dose compared to 51.2% in 2016. IPTp3 uptake was 43.0% in 2017 compared to 36.7% in 2016. In 2017, 22.1% received IPTp4 and 8.9% received IPTp5 [22]. These reports raise concerns that despite the policy change and progress made in implementation, the gap between high ANC attendance and IPTp-SP uptake remains significant [17]. Additionally, Ghana still experiences malaria and anemia in pregnancy, low birth weight among others [35–37]. These developments are worrying as they suggest that the investment in policy and resources to attain the minimum of 5 doses or more for all pregnant women are only achieving marginal gains over the years. The questions are: how effective is the ANC in delivering IPTp-SP; how are socio-cultural factors influencing IPTp-SP uptake and how can the ANC and communities be supported to ensure that the country's investment in IPTp-SP yields the desired results at a much faster rate?

The causes of low IPTp-SP uptake have been attributed to frequent stock-outs of SP, due to procurement and supply chain challenges, resulting in delays in getting SP into the country, especially during 2012–2014 [17]. Additionally, the country's Central Medical Stores (CMS) was gutted by a fire that destroyed the national SP stock in January 2015 [17]. Other studies have suggested that unemployment resulting in some women's inability to afford the cost of maternal healthcare service and late first ANC visit accounted for the low coverage of subsequent doses [1, 32, 33, 38].

This current study is part of the first author's (MA) post-doctoral research study on health system and socio-cultural factors influencing knowledge, attitudes and acceptance of malaria interventions among pregnant women in Ghana [39]. An earlier paper from the said research study Aberese-Ako, Magnussen [38] reported that healthcare managers dealt with stock-outs, government's failure to reimburse facilities for the cost of fee-free maternal healthcare and to supply health facilities with health products such as maternity record booklets by charging fees and rationing SP to women. Such managerial decisions contributed to the increasing price of SP and the general cost of ANC services. So women devised coping strategies such as skipping scheduled ANC appointments and laboratory tests, which compromised with quality health care provision. This current ethnographic study contributes to literature by exploring and describing factors that influenced uptake of IPTp-SP in healthcare facilities in Ghana by focusing on the daily interactions between healthcare providers and women as well as the socio-cultural context of healthcare provision.

## Methods

### Study design

The study design was ethnographic. It included non-participant observations, case studies, informal conversations and in depth interviews (IDIs) (semi-structured interview guides were used to conduct IDIs), to obtain data from healthcare providers, healthcare managers, pregnant women and community members. Informal conversation in this study is defined as: "*An unplanned and unanticipated interaction between an interviewer and a respondent that occurs naturally during the course of fieldwork observation. It is the most open-ended form of interviewing*" [40]. IDIs were more formal compared to informal conversations, as research assistants used semi-structured interview guides written with probes, transitions and follow-up questions, which provided more data, direction and control than the informal conversations [41:224].

Data were collected from April 2018 to March 2019. The research team comprised of a female medical anthropologist (MA) and 9 graduate research assistants (RAs). Three of the RAs were females and six were males, who could also speak the indigenous language of their assigned study areas: the Twi language for RAs who were recruited in the Ashanti Region and

the Ewe language for those who were recruited in the Volta Region. RAs observed and documented ANC care provision in the 8 study facilities and 8 communities. To prevent a Hawthorne effect, observations were conducted intermittently in the eight facilities and 8 communities [42]. MA trained RAs on observations and writing observation notes in accordance with Emerson, Fretz [43]. They were also trained to carry out community entry, to conduct informal conversations and IDIs prior to data collection and during the data collection process.

## Selection of research area

The study was conducted in five districts, three in the Ashanti region (the third district was included in the study, because the district was a new district that had been separated from one of the selected districts, so some of the pregnant women from the chosen district preferred to visit that health facility located in the new district) and two in the Volta region of Ghana. Eight health facilities (Table 1) and 8 communities were chosen for the study. Ashanti region was selected to represent the middle belt of the country, while Volta region was selected to represent the southernmost belt of Ghana. The two regions are linguistically different, Twi is spoken in the Ashanti region and Ewe is spoken in the Volta region. Ashanti Region reported the second highest percentage (98.8%) of women receiving ANC care from skilled providers in 2014, while the Volta region reported the second lowest percentage (93.9%) of women receiving ANC from skilled providers [29]. The district hospitals in the five districts qualified automatically to participate in the study. Also, interactions and interviews with pregnant women in some of the study communities revealed that they preferred to visit particular health facilities for ANC services. Three of such facilities, which are faith-based were included in the study. Thus, a total of 8 health facilities were selected for the study. Some women preferred the three facilities (2 in the Ashanti region and 1 in the Volta region), because they were closer to their communities than the district hospitals. The women's assertion of nearness to facilities was further confirmed when the study team conducted transect walk in all the study communities, to confirm the location of health facilities [38]. The study team visited the 8 health facilities and went through ANC records and maternity admission records for malaria in pregnancy (MiP) cases. The total number of MiP cases from January 2015 to March 2018 for the different

**Table 1. Study health facilities and communities in the Ashanti and Volta Regions with pseudonyms.**

| Type of study site | Region | |
|---|---|---|
| | Ashanti* | Volta# |
| | No. | No |
| Hospital(s) | 3 | 2 |
| Health Centre(s) | 1 | 2 |
| **Ownership of facility** | | |
| Government owned | 2 | 3 |
| Mission owned | 2 | 1 |
| **Communities** | 4 | 4 |

*Study facilities in the Ashanti region have been given the following pseudonyms: ASF01, ASF02, ASF03 and ASF04. Study communities in the Ashanti region have been given the following pseudonyms: ASC01, ASC02, ASC03 and ASC04.
#Study facilities in the Volta region have been given the following pseudonyms VRF01, VRF02, VRF03, and VRF04. Study communities in the Volta Region have been given the following pseudonyms: VRC01, VRC02, VRC03 and VRC04.

communities that access the services of each facility were tallied. The community with the highest total number of malaria in pregnancy cases in each facility was chosen to participate in the study. The average population for each study community was 10,000 inhabitants.

The study team conducted community entry activities such as visiting assembly members and chiefs and holding meetings with a cross section of opinion leaders to inform and to seek their permission to conduct the study in their communities.

## Selection of study participants

A research assistant was assigned to one health facility to carry out non-participant observation and to interact with health providers and pregnant women attending ANC. Convenience sampling was used to select pregnant women for conversations [41:27]. RAs took the phone number of any pregnant woman who was attending ANC and was willing to participate in an IDI. The woman was contacted later on and arrangement was made to meet her at her preferred venue for an in depth interview. The snowball method was also used to recruit pregnant women from the 8 study communities [40:115]. The first pregnant woman who was recruited helped the RA to identify other pregnant women in the community. The study was explained to them and those who were interested were recruited to participate in IDIs, after a written consent had been obtained. Opinion leaders such as assembly members, mothers and mothers in-law of pregnant women were invited to participate in IDIs.

Case studies were purposively selected from women who regularly attended ANC every month and those who were irregular or skipped ANC appointments. A total of 12 case studies were followed throughout the study period (Table 2). They were visited several times at home, where RAs observed how they took their medications, whether they honoured their ANC appointments, their experiences from their previous ANC visits especially on being offered SP,

**Table 2. Data collection methods and categories of respondents.**

| Region | Category of Respondents | IDIs | Conversations | Case studies | ANC interactions*# |
|---|---|---|---|---|---|
| **Ashanti*** | Health Managers | 8 | 4 | 0 | - |
| | Health Care providers | 11 | 20 | 0 | - |
| | Pregnant women | 30 | 25 | 4 | 40 |
| | Opinion Leaders | 10 | 5 | 0 | - |
| | Procurement officers | 1 | 2 | 0 | - |
| | Laboratory officials | 0 | 2 | 0 | - |
| | **DHD** officials** | **0** | **2** | **0** | - |
| | **Total** | **60** | **60** | **4** | **40** |
| **Volta #** | | | | | |
| | Health Managers | 8 | 4 | 0 | - |
| | Health Care providers | 12 | 20 | 0 | - |
| | Pregnant women | 40 | 32 | 8 | 40 |
| | Opinion Leaders | 14 | 6 | 0 | - |
| | DHD Officials | 0 | 2 | 0 | - |
| | **Total** | **74** | **64** | **8** | **40** |

*Observations were carried out intermittently in 4 health facilities and 4 communities from May, 2018 to March 2019 in the Ashanti Region.

** District Health Directorate.

#Observations were carried out in 4 health facilities and 4 communities from April 2018 to March 2019 in the Volta Region.

*# Eighty ANC interactions between health providers and clients were observed: 40 in the Ashanti region and 40 in the Volta region. An average of 10 were observed in each of the four facilities in each region.

and whether they were using LLINs. Also, their maternity record booklets were reviewed to confirm the information.

Health providers, mostly midwives and nurses providing ANC service, who had one year or more work experience in a health facility were selected to participate in the study. ANC unit managers (commonly referred to as in-charge), facility managers such as senior medical officers, physician assistants and administrators were interviewed to help understand managerial and administrative issues. The study team carried out follow-up informal conversations and interviews with procurement officers, laboratory personnel and officials at the district health directorate. The aim was to clarify some of the issues raised in IDIs and conversations with health providers and health managers. Details of the different category of study participants and the methods used for data collection are presented in Table 2.

## Data collection techniques and data collection process

An RA spent several months in a facility observing ANC procedures, interactions between healthcare providers and women who were attending ANC. RAs first observed women and health workers during the following ANC activities: registration of women, checking of women's blood pressure and protein in their urine, women being attended to in the ANC consulting room, women visiting the laboratory and the pharmacy.

In order to understand and experience the various processes that the women went through, RAs also selected ANC attendants at random and accompanied them throughout the ANC process. The RAs obtained permission from the health providers to interact with the women and they also sought verbal consent from such women to accompany them through the ANC procedures. The RAs talked to the women who they chose at random to clarify actions and activities that were observed. RAs wrote down notes on the conversations that they had with the ANC clients and the health workers and later typed them out. Observations and conversations with women focused on women's knowledge on SP, knowledge on MiP, their intention to take SP among others. Conversations with health providers centred on SP policies, SP availability, information offered to women before and after offering them SP. Conversations and IDIs that were conducted with pregnant women and community members were in the Ewe language for those in the Volta region and the Twi language for those in the Ashanti region.

The IDIs centred on knowledge, attitudes, beliefs and practices on malaria in pregnancy (MiP) interventions and socio-cultural practices. RAs conducted IDIs with healthcare providers and healthcare managers in English and they focused on maternal and MiP policies and service provision, challenges and facilitators. IDIs were recorded using digital recorders and they were transcribed verbatim to preserve interviewees' original messages and experiences. Interviews in Ewe and Twi were transcribed into English to enable easy analysis and comparison. The study used English language to conduct IDIs and conversations with healthcare providers and NHIS officials, because English is the official language of Ghana (see additional files for IDI guides and observation checklists).

Also, RAs obtained permission from the women to go through their maternity booklets to confirm IPTp-SP uptake.

## Data analysis

IDI transcripts, observation notes and notes from conversations were uploaded onto qualitative analysis software NVivo Version 11 to support the analysis. The data was triangulated and a coding list on common themes that arose from the data (IDIs, observation notes and conversations) was generated. MA and ED (ED is a qualitative expert who was hired to support coding of the data in order to enhance validity) independently coded the data thematically. The

analysis aimed at identifying similarities, patterns, differences and contradictions in the information observed or presented by study participants [44]. Main themes that were identified from the analysis formed the basis for interpreting and reporting on study findings. This manuscript is part of the larger study mentioned in the introduction, so some of the findings have been reported in the earlier paper [38].

### Ethics statement

Ethical clearance was obtained from the University of Health and Allied Sciences' Research Ethics Committee [UHAS-REC/A.I Ul 17, 18]. Written consent was obtained from all interview participants. Verbal consent was obtained from study participants that informal conversations were held with and for observations. While written consent is recommended for study participants, verbal consent can be used in situations where time is of the essence, as was the case with the informal conversations that the study team conducted with some of the clients attending ANC, who did not have time to participate in IDIs [45]

In this study women who were attending ANC were invited to participate in conversations and interviews. Those who had ample time for an interview were given time to reflect and their phone numbers were taken by the RAs. They were called at a later date by the RAs and if they consented, the RAs followed up to their homes for interviews, after they had obtained written consent from them. However, RAs conversed with women who were willing to participate in the study, but did not have enough time to participant in IDIs. For such study participants RAs first sought permission from ANC department heads and subsequently from the clients, who granted verbal consent to participate in the study. A few of those who were approached declined to be interviewed. Only one study participant was 16 years old and permission was sought from her mother prior to her inclusion in the study.

Permission to conduct the study was sought from district directors of health of participating districts, facility managers of the eight study facilities, department managers and chiefs and assembly members in the study communities. Besides actual country and region names, pseudonyms have been used for districts, individuals and facilities' names, to protect informants' identity. Health facility pseudonyms beginning with ASF refer to study facilities in the Ashanti Region and ASC refer to study communities in the Ashanti Region. While the prefix VRF refer to facilities in the Volta region and VRC refer to study communities in the Volta Region. Pseudonyms of respondents are thus predicated by the prefix of the facility or community that the observation, conversation and IDI was conducted respectively.

## Results

Findings from the study reveal that all the health facilities observed DOT, however most women did not take the optimum number of IPTp-SP doses during pregnancy. Three groups of factors that contributed to women completing 5 doses of IPTp-SP5 were health system, socio-cultural and individual. The details of these three group of factors are discussed in subsequent sections.

### Health system factors

Health system factors such as how ANC services were organized, strategies employed by health workers to administer SP and high trust in the healthcare system facilitated uptake of the first dose of IPTp-SP. However, healthcare providers providing inadequate information to ANC clients, facilities experiencing occasional stock-outs and charging fees for SP and other ANC services reduced uptake and adherence to subsequent IPTp-SP doses.

**Organisational arrangement facilitating IPTp-SP3+ uptake.**    Interviews and conversations with health providers at ANC clinics revealed that they knew that the first dose of SP is to

be given to women at 16 weeks of pregnancy. However, respondents presented varying opinions on the number of doses that a pregnant woman should receive during pregnancy. Three in five health providers said that the guideline indicated that women should receive 5 doses of SP, given monthly with the first dose at 16 weeks of pregnancy and the last at 36 weeks. Nevertheless, 2 in 5 health providers said that SP can be extended beyond 36 weeks, so a woman could be given 6 or 7 doses prior to delivery.

All eight facilities mostly practiced focused ANC and implemented the WHO's recommendation of administering IPTp-SP under DOT. In most facilities ANC procedures started with checking women's blood pressure and testing for protein in urine in an open space. In six facilities, the next set of procedures were conducted in a consulting room. The women were examined to confirm that their gestational age was 16 weeks or more, either through a scan or through physical examination. They were offered seats opposite to the midwives' position, which enabled midwives and clients to have a face-to-face interaction. Midwives usually kept a box of SP on the consulting table. They removed a pack, ripped the three tablets from the package, placed them in the right palm of the women and watched them chew or swallow them (Table 3). Afterwards, the healthcare providers recorded the uptake into the women's maternity booklet before returning it to them.

Each facility had an ANC outpatient department (ANC OPD), where all the pregnant women converged to be called one at a time into an ANC consulting room. Usually, when the women appeared for ANC, each woman placed her booklet on a table located in front of the ANC OPD. A nurse or a ward aid (nurse assistant) went through the booklets to determine those who were due for SP (VRF01, observation notes, 24/07/2018; ASF04, Observation notes, 20/08/2018). The booklets were then given back to the women after the vitals had been checked and recorded. The study team observed that in the ANC consulting room a health provider's first approach was to take the booklet from the ANC attendee to read her records to know her history of SP uptake and other treatment, to guide the health provider in clinical decision making (ASF01, Observation notes, 29/08/2018; ASF02, Observation notes, 27/08/2018; ASF04, Observation notes, 20/08/2018; VRF01, Observation notes, 23/07/2019; VRF03, Observation notes, 04/09/2018).

All the pregnant women that were observed in the 8 facilities had a copy of the maternity booklet or were in the process of receiving one. Each pregnant woman was given one copy of the maternity booklet when she visited the ANC for the first time and tested positive for pregnancy (ASF01, Observation notes, 29/08/2018; ASF02, Observation notes, 27/08/2018; VRF01, observation notes, 23/07/2018; VRF03, observation notes, 04/09/2018). All the different components of care given to her including the issuing of each dose of IPTp-SP was recorded in the maternity booklet. She was required to carry the booklet along with her on each visit. The health provider referred to the book on each visit to determine the woman's healthcare history and whether she was eligible for the next SP. A midwife explained that the booklet is accepted in all facilities nationwide, so women carry it even when they travel outside their catchment areas, in order to enable them access ANC at any other facility in the country (ASF04, IDI, Health worker, 02).

Prior to the start of ANC procedures in VRF01, a health provider called out names of all women who were due to take SP on that day. In ASF03 and VRF01 a midwife provided SP in the open to all women who qualified to take SP. She first offered them a seat and then she offered them three tablets of SP and watched them swallow or chew it.

Facility ASF02 and VRF02 provided water to ANC clients, while ASF01 and VRF04 sold water to them. In facilities that did not provide water to clients, health providers took the women's maternity booklets, asked the clients to go out to get water and return to take the SP

**Table 3. Experiences of IDI participants in taking SP under DOT.**

| Themes | Reponses obtained on the different themes from pregnant women |
|---|---|
| **Whether women were offered SP during their last visit to the ANC** | *"Yes, medicine for malaria [SP] and you drink it at the hospital. They won't allow you to bring it home, because it is very difficult to drink it. I nearly vomit after drinking it."* (ASC04, IID, pregnant woman 03) |
| | *"And we are also given some 3 white tablets to chew in their presence."* (VRF02, IDI, pregnant woman 04) |
| | *"...when I went, they gave me that "three-three" [SP]. They made me take it in front of them."* (VRC02, IDI, pregnant woman, 04) |
| | *"We take it in front of the nurses. Some people when they bring it home they won't take it and they will throw it away. It is very difficult to take it, so they will tell you to take it there."* (ASC03, IDI, pregnant woman, 07) |
| | *"We are given a certain malaria drug that we take inside the consulting room, before getting out."* (VRC03, IDI, pregnant woman03) |
| | *"Yes, you drink it there. They don't allow you to take it outside. The midwife takes the drugs out of its cover and put them in your palm and you drink it with water."* (ASC03, IDI, pregnant woman, 01) |
| | *"...you will have to chew and swallow it in their presence, before leaving."* (VRC03, IDI, pregnant woman, 08) |
| | *"That one [SP] when you get to the nurses, they will give you to take before you go out."* (VRC01, IDI, pregnant woman, 01) |
| | *"Yes, the first time I went to ANC, it [SP] was given to me and I took it right there, in front of the nurse."* (VRC02, IDI, Pregnant woman, 03) |
| **Women's experiences of being asked to bring water to take SP** | *"They ask us to bring water. So when they put it in your palm, then you have to chew it and swallow it in their presence."* (VRC02, IDI, 02) |
| | *"It is a tablet, when they [nurses] give you that medicine, they will ask you to go and buy sachet water and come. So when you come, they [nurses] will take it out of the wrapper and give you to take, to prevent in case malaria want to inflict the unborn child."* (VRC01, IDI, pregnant woman, 06) |
| **Experiences of women on being asked to eat before returning to take SP under DOT** | *"They [nurses] asked me if I had eaten and I said no. So they asked me to go and eat and come, after which I was given the SP. I just swallowed it like any other medicine. I took it in front of them."* (VRC01, IDI, pregnant woman, 07) |
| | *"They will ask you if you have eaten. If you have not eaten, they will ask you to go and eat and come for the drug. Then they will give you to take."* (VRC01, IDI, pregnant woman, 09) |
| | *"They ask you whether you have eaten or not. If you have not eaten, they tell you to go find something to eat and come back for the drug. They say the drug is a strong drug."* (ASC03, IDI, pregnant women, 09) |
| | *"There are three tablets that are given at once for us to swallow; but when you take it and you do not eat, it will disturb you."* (VRC03, IDI, Pregnant woman, 07) |
| **Women who were given some information on the relevance of SP before or after intake of SP** | *"They said it [SP] is for malaria. I also saw that they drew mosquito on the medication, so I knew it was a malaria drug."* (VRC04, IDI, pregnant woman, 03) |
| | *"...they said it is for fever or malaria."* (VRC04, IDI, pregnant woman, 06) |
| | *"They said it prevents us from getting malaria. It also protects the unborn child from getting malaria."* (ASC04, IDI, pregnant woman, 10) |
| | *"The midwife told me the drug [SP] will protect my unborn child and me from malaria."* (ASF04, conversation with an ANC client, 16/08/2018). |
| | *"They said for our unborn child to be healthy"* (VRC01, IDI, Pregnant woman 01) |
| **Women who were not given information on the relevance of DOT** | *"No, they have not told us anything [about SP], and we have not asked anything about it ourselves. But I know it is a malaria drug, so I have not asked anything about it."* (VRC03, IDI, Pregnant woman, 07) |
| | *"They didn't tell us anything about it [SP]. We just have to take it."* (VRC02, IDI, pregnant woman, 05) |
| | Researcher: *"They say it is what kind of drug?"* |
| | Respondent: *"They've written SP or something..."* |
| | Researcher: *"SP...and they say what does that drug do?"* |
| | Respondent: *"I haven't asked that intentionally."* (ASC01, IDI, 01) |

tablets in front of them. The health providers then recorded the intake in the women's maternity booklets and the women left to continue with the rest of the ANC procedures.

Most health providers explained that SP was 'strong', so they ensured that women ate before taking it, as a way of addressing complaints from women that they experienced side-effects of

SP. Health providers considered food such as porridge or tea as too light. So women who mentioned that they had eaten such foods were asked to find heavy food such as banku, kenkey (solid Ghanaian meals made from maize) to eat, before returning to take SP (Table 3).

In depth interviews with pregnant women and health providers, conversations and observations in ANC clinics revealed that one in two women were not given information on the relevance of SP, the number of doses that they were supposed to take during pregnancy and when to return for the next dose (Tables 3 and 4). However, it was observed that health workers usually wrote the date of the next scheduled ANC visit (which was usually one month from the current visit) in each client's maternity booklet (Table 4). Health providers were observed to give detailed information on SP to clients who were refusing to take SP and in most cases they were successful in convincing them. However, health providers reported that education was critical for adherence, so they gave the women ample education. Nevertheless, a few health workers admitted that they failed to give some of the women relevant information on SP, because of the heavy client load that they had to cope with.

Sometimes, women who were due for SP were diagnosed of having malaria parasites, which was usually confirmed after a laboratory test was conducted. Such a clinical decision was taken because, some women appeared at the ANC with complaints akin to malaria infection. In some instances women who were taking SP for the first time were made to undertake a test for malaria even if they did not show or complain of any symptoms. Those who tested positive were treated for malaria, but were not offered SP. Such women were informed that they would receive SP on their next visit to the health facility, if the malaria parasites were cleared. In some facilities such clients were asked to visit the laboratory to test for malaria parasites on the next ANC visit and if they tested negative, they were given SP. It was observed that in ASF03 clients who were previously treated for malaria were given SP on their next scheduled visit without being tested for malaria parasites. A health worker explained: "*We assume that the person has been treated after a month, so we start giving them the SP.*" (ASF03, Conversation with a health provider, 28/08/2018)

**Community health nurses' role in SP uptake.** Facility VRF02 reported that in order to increase uptake of SP among women community health nurses provided information on ITNs and SP to community members and households during community outreach programmes in the district. This initiative was to help to create awareness among community members and to encourage women to attend ANC in order to access ITN and IPTp-SP. Additionally, some of the community health nurses administered subsequent doses of SP to women during community outreach programmes. The community health workers were guided by documentation of the first dose of SP in the women's maternity booklets. The practice was to ensure that the first dose of SP was always administered at the ANC clinic and recorded in the patient's maternity booklet, so that anywhere she went within Ghana a health provider could refer to the booklet, to make informed clinical decisions including offering her subsequent doses of SP. Thus, the subsequent doses could then be offered in the communities by the community health workers who visited the communities frequently for health outreach programmes. This ensured that women who were reluctant to honour subsequent ANC appointments, those who did not have money to visit the ANC and those who complained of long distance to the ANC could still get SP.

**Trust in the health care system and belief that taking SP was compulsory influenced uptake.** Clients' trust in the health care system and health providers motivated them to take IPTp-SP. It was observed that ANC clients trusted health providers, so they took in SP that was given to them by the providers even when they were not given information. They believed that health providers were concerned and interested in their welfare, so they will not mislead or harm them. Additionally, health providers continued to tell women that SP was

**Table 4. Observations in 8 ANC clinics on SP intake under DOT.**

| Facility code | Observations in 8 facilities |
|---|---|
| | **Ashanti Region** |
| ASF01 | Client 01: The midwife told her she was going to take medication, which will protect her and the pregnancy from malaria. The midwife brought out SP and gave it to her, together with water and the woman took in the medication. (ASF01, observation notes, 29/08/2018) |
| | Client 02: The midwife asked her if she had any complaints, the woman said she had none. The midwife asked her if she had eaten before coming and the woman said yes, she had. The midwife gave her SP. It was her 3rd dose of SP. She was asked to return in a month's time. (ASF01, observation notes, 29/08/2018). |
| ASF02 | Client 01: The midwife told her she was going to take medication. The midwife brought the SP out and gave it to her, together with water and the woman took in the medication. (ASF02, observation notes, 27/08/2018) |
| | Client 02: The midwife asked her if she had eaten. The woman replied that she hadn't. So the midwife asked her to go and eat, because she will take medication. She went and came back in some few minutes and the midwife took the SP out and gave it to her together with water. After the woman had taken the medication, the midwife asked the woman to go and her next visit was scheduled for 28th September, 2018. (ASF02, observation notes, 27/07/2018) |
| ASF03 | Client 01 is seven months pregnant. After bringing her urine for testing, she was given SP, which she was given sachet water to take in the full glare of everybody. (ASF03, observation notes, 20/08/2018) |
| | Client 02, a seven months pregnant woman gave her urine to the in-charge for protein/glucose test. After the test she was given SP to take. Client 01 kept the SP in her hand bag. The nurse who gave it to her realizing that screamed at her to bring the SP out and added: "*When you go to the house you won't take it*" Client 01 replied: "*I will take it just that I feel dizzy when I take it*". One of the nurses gave a sachet of water to client 01 and she took the SP there, in front of the nurses. (ASF03, observation notes, 03/09/2018) |
| | Client 03 is sixteen years old. This happens to be first pregnancy. When she took the seat, the nurse took her vital statistics, tested her urine and asked her "Have you eaten?" She said no. The nurse then told her to take the drug and after that go and find something to eat. She gave the drug to Client 03 and she peeled it out of the pack and took it with water. (ASF03, observation notes, 10/09/2018) |
| | Client 04, the midwife told Client 03 that she needed to do some labs for her to ascertain how she is doing after the treatment given to her by the doctor. The midwife asked her to go and buy water and come for the SP. She came after a short while with the water and took the SP in the presence of the midwife. (VRF03, Observation notes, 08/10/2018) |
| ASF04 | Client 01 went back to the maternity ward, where she was given SP. It was observed that water was not provided for her. So she went to the entrance of the facility, bought water and returned to the ANC and took the SP in the presence of the midwife in-charge. She paid three (3) cedis for the SP. (ASF04, observation notes, 30/08/2019) |
| | Client 02: SP was served on DOT, but no education was given because it is her third dose of SP. The client paid 3cedis for the SP served. (ASF04, observation notes, 15/08/2018) |
| | **Volta Region** |
| VRF01 | Before SP was given to a woman she was asked whether she had eaten. If she responded in the affirmative, they asked her to bring water to take the drug in their (nurses) presence. If she responded in the negative, she was told to go and eat, get some water and come back for the drug. Some chewed and drank water afterwards, others swallowed the three tablets, one after the other with water, while others swallowed the three tablets at once with water (VRF01, Observation notes, 24/07/2018) |
| | Client 01: At the SP table, client 01 pleaded that her SP be given to her to send outside, get water and take it there at once, but the nurses refused. She made all efforts to convince them to trust her, but to no avail. So she had to leave her maternity booklet on the ANC table and go to get water. She returned with water to take the drug in the presence of the nurses. (VRF01, Observation notes, 1/08/2018) |
| VRF02 | There was a big veronica bucket with water and several drinking cups placed on top. Each client was told to fetch water in a cup and come for the SP. (VRF02, observation notes, 29/06/2018). |
| | Client 01: After Client 01 took the SP in the consulting room, she was then directed to the pharmacy for her routine drugs (VRF02, observation notes, 23/10/2018). |
| | Client 02 left the ANC consulting room to the lab and came back in an hour with the result in her ANC booklet indicating, 'No malaria parasites'. The midwife then asked her to fetch water and come for the SP. She fetched water from the veronica bucket and went to take the SP from the midwife and took it in the presence of the midwife (VRF02, 25/10/2018). |
| VRF03 | Client 01 came into the consulting room. . .The midwife asked her if she had any issue for today. Client 01 stated that sometimes she felt abdominal pains. The midwife explained the reason for the abdominal pain and after a series of interactions, the midwife told her that they had a malaria drug that they give to pregnant women, starting from the fourth month of pregnancy and it protects the mother against malaria, aside sleeping under the bed net. The midwife asked client 02 to go and buy water and come for the SP. Client 02 went out and returned after a short while with water. The midwife gave her SP, which she took in the consulting room. (VRF03, observation notes, 08/10/2018) |
| VRF04 | The midwife at the ANC had a bag of water. So when pregnant women came for ANC and were given SP, they were told to buy water at 20 Ghana pesewas, to take the SP. The clients did not complain or say anything, but just bought the water and took in the SP. (VRF04, observation notes, 27/06/2018) |
| | Client 01: The midwife gave Client 01 SP to take. The midwife didn't say anything to her before giving her the SP. Client 01 seemed to know the norm. She stood up and placed 20pesewas on the table and picked a sachet of water and came back to sit down. She collected the SP from the midwife and took it. The midwife told Client 01 that she should make sure she eats well and takes in her routine drugs (VRF04, observation notes, 30/07/2018). |
| | Client 02's first visit was on 26/07/2018. Today (22/8/2018), was her second visit. After her urine was tested, she was asked to buy water to take SP. Before she was given the SP, the midwife said (holding the SP), "*This medicine is to prevent you from getting malaria*.' She nodded and took the SP. After she took the SP the in-charge did not tell her anything. (VRF04, observation notes, 22/08/2018) |
| | Client 03: All the pregnant women who came for ANC at the facility were not spoken to before or after they were given SP, except for Client 03 who tested positive for malaria parasites. . . At a point the midwife pointed at the SP box, which was lying on the table and said to Client 03: "*This medicine is to prevent you from getting malaria, but when you expose yourself to mosquitos you will get malaria.*" (VRC04, observation, 08/08/2018) |
| | Client 04 came in for ANC for the first time. She was given an ANC booklet and a mosquito net by the ANC in-charge. Her name was written in the ANC records book. The in-charge told her that on her next visit she would be given SP, so she should do well to come. (VRF04, observation notes, 08/08/2018) |

compulsory. Thus, some women believed the health workers' claim and felt that they had no choice but to take SP, even though some were not told by the health workers the reason why it was compulsory.

Some women were motivated to attend ANC regularly, because they had high interest in protecting themselves and their babies, and they trusted that the health care system could

provide them such protection. Such women trusted all the information given to them at ANC and were always willing to comply with directives including fulfilling all the scheduled ANC visits and taking IPTp-SP. The following quotations illustrate experiences of clients:

"*Some of the medicine [SP], we take it there. Some people don't take it when they bring it home, so they give you water and tell you to take it. Some people don't like taking medicines and if you do that you are causing harm to your child. They* [health providers] *want the best for you, so that everything will be okay for you and your child.*"

(ASC04, IDI, pregnant woman 02)

"*. . . there are some people who will collect it [SP], but will not go and take it. So they (midwives) are also responsible for protecting us against malaria and for preventing the child from getting it. So, they ensure that we take it in their presence.*"

(VRC03, IDI, pregnant woman, 09)

"*For the malaria drug (SP) everybody must take it.*"

(ASC03, pregnant woman, 07)

"F*our times [taken SP], and when I take that medicine [SP], I feel dizzy. It's difficult to take this particular drug [SP], but because I don't have any option, I have to take it.*"

(ASF04, conversation with an ANC, 16/07/2018)

**Effect of stock-outs on IPTp-SP3+ uptake.** Occasional stock-outs of SP in health facilities affected uptake of optimum doses of IPTp-SP. Some facilities dealt with stock-outs by prescribing SP for ANC clients to buy from private pharmacies (most of the private pharmacies are a walking distance from the health facilities). Sometimes, women's maternity booklets were seized and only given back, when they returned to the facility to take the medicine under DOT:

"*Because they take the SP under direct observed therapy, when we don't have the SP in the hospital, we write it and give it to the pregnant women to buy the SP outside the hospital. After they have bought the SP, they will then bring it to the facility, to be taken in our [midwives] presence. We seize their ANC booklets. So after they have bought the SP outside the hospital and have taken the SP in our presence, then we give their ANC booklets back to them.*"

(ASF02, health provider, 02)

Three facilities [ASF02, ASF03 and ASF04] had regular supply of SP, because they the solved stock-out problem by purchasing SP and selling it to clients. They sold it as a package with routine drugs or haematenics (such as folic acid, multivite and ferrous sulphate) at 3GH₵ ($0.6). However, some clients could not afford the price of the package of drugs, so sometimes they skipped scheduled ANC appointments and missed some doses of SP. A woman who was nine months pregnant and attended ANC in facility ASF04 indicated that SP was beneficial to pregnant women, yet she had taken it only once. She failed to take the subsequent four doses, because she could not pay for the drug as the interaction between her and a research assistant illustrates:

RA: *"Have you ever been given this drug [showed her the SP] before?"*

Respondent: *"Yes they have given me this drug [SP] before and I took it at the hospital. They said I will take it five times before giving birth, but I have only taken it once"*

RA: *"Have you asked the nurses why they haven't given you the malaria drug again?"*

Respondent: *"No, I haven't asked them. One other thing is that the government does not cover it and even if the government covers it, he does not pay all. So, you have to buy it yourself and it costs 3GH₵ ($0.6). If you don't have money they will not give it to you."*

RA: *"You mean they sell the SP at the hospital?"*

Respondent: *"Yes, the first day I took it, I bought it for 3GH₵ ($0.6). So, if you don't have money to buy it, they won't give it to you."*

In some circumstances when facility ASF04 had limited quantity of SP in stock, it rationed SP to clients as a healthcare manager revealed:

*"With the IPTp-SP that is the malaria prophylaxis that they give to the pregnant women, which is supposed to be taken monthly—that is after 16 weeks of pregnancy, it is not readily available. So in theory the IPTp-SP is given monthly after quickening, but practically the midwives are still using the old system, where they give it to the pregnant women in two months interval, instead of one month interval and then stop giving it to the pregnant women at 36 weeks of pregnancy. It will be good if the SP could be made readily available, so that the pregnant women can take it monthly."*

(ASF04, IDI, Health manager, 03)

Facility VRF02 addressed stock-out of SP by borrowing SP from sister facilities for clients. A second strategy that facility VRF02 used to address stock-outs, which was also practiced by facility VRF04 was to ask clients to return at a later date for SP. Clients returned when SP was available to get their regular dose.

## Socio-cultural factors influencing IPTp-SP uptake

Socio-cultural factors such as women being encouraged by family members and friends facilitated initial and continued uptake of IPTp-SP. However, women's inability to afford the cost of SP, social commitment and household decision making on ANC attendance hampered continued uptake of IPTp-SP. For young unmarried women, health system and socio-cultural factors influenced their use of ANC services, which affected uptake of IPTp-SP.

**Interpersonal relations, social commitment and ability to afford SP influenced IPTp-SP uptake.** A husband's ability to pay for the cost of ANC care and to advise or force his wife to attend ANC, influenced ANC attendance. An opinion leader stated: '*We also tell them [husbands] to force their wives to go for ANC.*' (VRC04, IDI, Assembly member)

Friends, family members such as sisters, mothers and mothers in-law encouraged women to attend ANC and to take SP, which contributed to uptake. In some cases family members especially mothers and mothers-in-law decided on when a woman should begin ANC. For instance a husband who accompanied his wife, who was 4 months pregnant to the ANC, explained that his mother and his mother in-law decided that his wife should attend ANC at 4 months. This is because, his mother and his mother in-law had a history of attending ANC in their fourth and fifth months of pregnancy respectively (ASF02, conversation with the husband of an ANC client, 22/08/2018). Others on the other hand were encouraged to attend

ANC by their friends (ASC01, conversation with a case study at her home, 10/01/2019; ASF02, conversation with an ANC client 22/08/2018)).

Some pregnant women were discouraged by friends and female relatives, to go for ANC early, as they were told that they would have to pay fees.

Some women combined ANC visits with taking herbs, visiting herbalists and pastors for health care, which affected adherence to ANC directives including taking SP. In community ASC02 three in four pregnant women combined medicine given at the ANC with herbs. When they discovered that they were pregnant, they first went to an herbalist and delayed visits to the ANC, which affected early uptake of SP. They explained that the herbs helped to prevent the unborn child from getting '*Asiram*' [hydrocephalus]. They believed unborn babies are exposed to spiritual and physical attacks by enemies. Also, there were instances that pastors decided on when church members could start attending ANC and when they should eat during pregnancy. A midwife shared an experience with a client who was refusing to take SP: *"A pregnant woman. . . came here last time. 'Take the drug,' 'I am fasting!. . .The Pastor said I should fast.'. . .So I told her that I am Osofo Mame [wife of a pastor or a female pastor], but I won't ask a pregnant woman to fast. . ..So she went and ate and came and took the medicine* [SP]." (VRF01, IDI, Health Worker, 04).

In community ASC04, it was common for women to relocate from their husbands or their own residences to their mothers' places [mostly their mothers lived in different communities] to deliver. ANC providers could not document whether such women completed 5 or more doses of SP prior to delivery. The following quotation illustrates the challenges health workers face in such circumstances:

"*About 60% of the pregnant women who come here (ASF04) for their ANC service complete the first, second and third doses. Some of the pregnant women when they are about to deliver, they go to their mothers' houses, mostly outside* ASC04. *So, they usually take the fourth and fifth doses at different hospitals, that is if their parents stay outside* ASC04. *So we are unable to capture fully the number of pregnant women who take the fourth and fifth doses.*"

(ASF04, IDI, Health worker, 04)

Women were encouraged by health providers to carry their ANC booklets whenever they travelled. So that they could attend ANC on their scheduled dates at the closest health facility.

Some women skipped scheduled ANC appointments due to social commitments such as funerals, having to go to the market on their scheduled ANC dates and family responsibilities such as going to the farm. Thus, missing out on the monthly intake of SP. Nevertheless, it was observed in ANC clinics that some midwives encouraged clients to carry their ANC booklets along, whenever they traveled out of town. This was to enable women attend ANC clinics that were deemed closer during travels. ANC providers also encouraged clients to explain to their family members to allow them to attend ANC on their scheduled ANC appointment dates, if family members wanted them to engage in other activities on such days.

**Health system and socio-cultural factors influencing young women's access and uptake of IPTp-SP.** Majority of the unmarried young women between the ages of 18–24 years old, who the study team interacted with were unemployed, or had recently dropped out of school, because they had become pregnant and were not receiving support from their partners. For most of them the pregnancy was unplanned and unwanted, which affected their attitude towards utilizing ANC and MiP services including the uptake of IPTp-SP. Some of them were brought to the ANC by their mothers or grandmothers, sometimes in their sixth, seventh, eighth and even ninth month, so they did not receive the minimum 5 doses of SP. Table 5

**Table 5. Young women's experiences in seeking ANC care and IPTp-SP uptake.**

| Young women's experiences at the ANC |
| --- |
| *"Some pregnant women, they can stay at home for a very long time and report with the pregnancy very late. For example, one girl [adolescent] came today, the pregnancy was already six months. She was not even having health insurance, so she had to go and get the NHIS* (national health insurance scheme) *card and return. People like that are not able to take all the 5 doses before delivery."* (VRF02, IDI, Health worker, 03) |
| Adolescent client 01, was brought in for ANC for the first time in the ninth month of pregnancy by her mother. She was not given SP, as the midwives explained that she was due to deliver, so it was too late to start her on SP. (ASF02, Observation notes, 27/08/2018). |
| Adolescent client 02, a 23 year old second year pharmacy student, was seven months pregnant. She started attending ANC in her sixth month. She indicated that she would never take SP, because she did not like the smell and it causes her to vomit. She also admitted that she was not sleeping under a bed net and that she was not taking her routine ANC drugs. She lamented that the man who made her pregnant had absconded and her parents were peeved. She said if she had detected the pregnancy early, she would have aborted it. However, she only detected it in her fourth month, so she was afraid that if she aborted it she would die. From all indications she was not enthused about the pregnancy, so she had a negative attitude towards protecting herself from malaria and its negative consequences on the pregnancy (ASF02, Observation notes, 23/08/2018). |
| Adolescent client 3, was a 16 year old who had stopped attending school after Junior High School. She was impregnated by a boy who was in the Senior High School. She was about seven months pregnant when the research team visited her at home for the first times. She had never attended ANC, so she had never taken SP. She explained that the boy did not have money to support her and her mother did not also have money to support her that was why she had not started attending ANC (ASC03, visit to case study, 06/08/2018). |

presents interviews and observations of young women who sought ANC services. Most of the cases were observed in facility ASF02 in the Ashanti region, which was the only facility that devoted one day in a week to conduct adolescent ANC clinic.

Facility ASF02 addressed the challenge of getting adolescents to utilize ANC services by devoting one ANC day to adolescents. They served soda drinks to welcome them to the ANC and this motivated them to attend ANC. This practice ensured that adolescents attended ANC, which facilitated uptake of IPTp-SP.

Facility ASF01 reported that one day in a week was devoted to adolescents, so the team visited the facility on two occasions on the reported allotted day and time to observe the adolescents' clinic, however the team did not find any adolescents. So the team questioned the staff and no explanation was offered, thus our observations suggested that no day was actually set up for adolescent ANC. In facility VRF01 the study team observed on one occasion an adolescent who was attending ANC being reprimanded for mistakenly placing her urine bottle on an ANC table. Other pregnant women who made similar mistakes were not reprimanded (VRF01, Observation notes, 24/07/2018).

Also, a case study who was 17 year-old told the research team that she was unwilling to attend ANC at the nearest health centre, which happens to be VRF02, but rather travelled to a distant lower level facility, a community-based health planning and services (CHPS) compound, because she believed that the nurses did not treat her with respect at VRF02. She admitted that sometimes she could not afford the cost of transport to the distant facility, so she had skipped her latest ANC appointment as at the time of the team's visit to her home (VRC02, Conversation with a Case Study, 19/09/2018).

## Individual factors influencing IPTp-SP uptake

Individual factors such as women's knowledge of the relevance of SP, women's desire to protect themselves and their unborn babies motivated them to take SP. But some women refused to take SP, others skipped scheduled ANC appointments and some women started ANC late, which made it difficult for them to complete the recommended optimum doses of IPTp-SP.

**Individual factors contributing to low uptake of IPTp-SP.** Interviews and conversations with study participants revealed that there were instances that women declined to take SP, even when they were eligible. Some complained of vomiting and nausea, but others did not

have any tangible reason for declining to take SP. In both regions most refusals came from professional teachers. It was noted that health workers made efforts to persuade such women to take SP, but they were not successful.

> "*Teachers, they think they know. They read the 'Wikipedia thing'. So they will come and challenge you. So they normally refuse it* [SP]. . . *I had two cases. . . They said, 'That thing [SP], any time I take it I vomit the whole day, I won't take it.' I talked, talked, talked. I had to refer them to my second colleague, to advise them, but they refused. . . So I wrote it boldly there* [in their maternity booklets], *'Refused'".*
>
> (VRF01, IDI, Health worker 04)

> "*Almost all my clients take the SP. I have a client who is a teacher, so far she is the only one, who is refusing to take SP. So what I did was I recorded it in my book that she is refusing to take SP. Her lab test shows that she can take the SP, but she refused to take it. Like the saying goes, 'patient's right'. She is a teacher, so I assume she knows the right thing, but if she says she won't take it, I can't force her.*"
>
> (ASC02, Health provider 01)

Nevertheless, lack of appropriate information and lack of experience in taking SP contributed to refusal. An RA interacted with a woman who had completed Junior High School and was four months pregnant with her second child. Her responses suggested that she had never taken SP as the excerpts illustrate:

> RA: '*They say it is what kind of drug*?'
>
> Respondent: '*They have written SP or something. . .*'
>
> RA: '*SP. . .and they say what does that drug do*?'
>
> Respondent: '*I haven't asked that intentionally. . . Ok, when I got pregnant with my first child, I never took it. They said it was unavailable. . . So when I went yesterday they said they were going to give me some* [SP]. *However, I told them I had not eaten, so I was going to take some food and return, but I didn't take it.*'
>
> RA: "*Okay. So how did you manage to avoid taking it, such that they also didn't realize that you had not taken it*?"
>
> Respondent: "*I didn't use any means, I just told her* [health provider] *that I had not eaten. . . She said she was not letting me off the hook, simply because I had not eaten. She said from her estimation, my time had not advanced that much and that I would eventually take some. So, I should go and return later. Thus, I just went for my medications and left. I didn't take it.*"
>
> (ASC01, IDI, pregnant woman, 01).

Several pregnant women reported that they started attending ANC late, because they did not have money to pay for ANC services. Others started ANC early, but skipped some of their scheduled ANC appointments, as a way of coping with the cost, which was confirmed by a health care provider (ASF01, IDI, Healthcare provider 02). This reduced their chances of obtaining the monthly required doses in order to attain 5 or more doses by the time they delivered.

Late attendance to ANC contributed to late initiation of IPTp-SP and women not being able to complete optimum doses of SP. Also, after the third dose some women defaulted and

only returned to the facility to deliver, while others came to register for ANC, received a dose of SP and did not return until they were due to deliver. Health workers shared their experiences:

"*Some of the pregnant women don't complete all the 5 doses before delivery, because some of them don't visit the facility at the early stage of the pregnancy. They stay at home for long, so some of the women sometimes take only three doses of the SP before delivery.*"

(ASF04, IDI, Health worker, 04)

"*The uptake is Okay, it is encouraging. Only that the fourth and fifth doses are not encouraging, because they don't register early. Others also default.*"

(VRF02, IDI, Health worker, 01)

**Clients who took SP without being observed by health providers.**    During the study period, only three women (out of 70 IDIs, 57 conversations and 80 ANC observations), were observed or reported that they took SP without being observed by a health worker. In ASF04, it was observed that an ANC client who was pregnant for the first time took her first dose of SP outside the ANC consulting room unsupervised. When she was asked why, she explained that the midwife asked her to take the SP on her own, as the midwife had other issues to attend to (ASF04, observation notes, 26/09/2018). An ANC client, who was nine months pregnant in ASF03, reported that she took some of her SP doses at home unsupervised. She said she experienced dizziness and nausea when she took her first and second doses under DOT. So on her third visit the health providers agreed that she could take the subsequent doses at home (ASF03, Observation notes, 10/09/2018). Another respondent in community VRC03, reported that she bought her second dose of SP from a private pharmacy and took it unsupervised. She said she made the decision, because she wanted to protect herself from getting malaria.

**Women's knowledge on when to start taking SP, number of doses and willingness to take SP in future.**    Most women knew that they were supposed to take SP, including those who were not due for SP. However, majority of women were not sure when they were supposed to start taking SP and some said they had no idea of the number of times that they were supposed to take SP, because they were not told:

"*They told me the time to take it is not due. . ..I didn't ask them, but here they have a specific time that you take it. I am just 4 months* pregnant, *I haven't reached far.*"

(ASC04, IDI, pregnant woman, 05)

"*For that one* [number of times to take SP], *we were not told. So, we know nothing about it. Whenever you go and you are told to take medicine, then you go for it. So we do not know how many times it should be taken.*"

(VRC03, IDI, Pregnant woman, 07)

Women gave various responses on the number of doses of SP that they were required to take during the period of pregnancy. Most women indicated three, a few mentioned two, four and five, while some said they were not told when they asked the midwives. The quotations below testify to the different responses:

*"The last time I asked, they only told me I was supposed to take it four times. They did not give me any reason. I was only told it is four times, I should not talk much and that it is my duty to take it."*

(VRC03, IDI, pregnant woman, 03)

*"I have drunk it 2 times and it is left with 1 to drink. I will finish drinking mine this month."*

(ASC04, IDI, Pregnant woman, 03)

*"There were these 2 clients, when I was giving them the 4th dose, they said they had finished. I said 'oh, it was first that you take only three doses, now you have to take 5 doses.'"*

*(*VRF02, IDI, health worker, 02)

*"Yes, they have given me this drug [an RA showed her the SP] before and I took it at the hospital. They said I will take it five times before delivery, but I have only taken it once and my date of delivery is almost due."*

(ASF04, conversation with ANC client, 16/08/2018)

All women who the study team interacted with including those who complained of side-effects such as feeling dizzy and weakness, said that they were willing to take SP in the near future. Only one woman said she was not willing to take future doses.

## Recommendations for promoting IPTp-SP uptake

Healthcare providers provided recommendations on how IPTp-SP uptake can be improved. Recommendations included giving health education such as encouraging women to start ANC early, stressing the importance of SP to women and motivating them to take it. Some health providers suggested that SP should be made readily available at health facilities. An ANC manager stated: "*The government should provide the SP and it should be given to us frequently. They should not let us experience stock-outs of SP.*" (ASF02, IDI, ANC manager)

Healthcare providers in facility VRF02 recommended that community health nurses should be encouraged to include community education on regular ANC attendance and SP uptake during outreach programmes. Also, community health nurses should be encouraged to administer subsequent doses of SP to women who are due in communities. This will save the women time and the inconvenience of having to travel to a health facility for subsequent doses.

ANC clients recommended that they should be given adequate information on how to prepare well before taking SP and on the side-effects. A case study who the study team visited at home said that she prefers that it is put in liquid form (ASC04, conversation with case study, 10/04/2019).

Opinion leaders such as mother in-laws, assembly members and community elders recommended community education and husbands' involvement in encouraging their wives to attend ANC and to adhere to all treatment offered.

## Discussion

This study used ethnographic research methods to explore how health system factors such as organisational arrangements, health workers' strategic approach to enforcing DOT and trust in health facilities facilitated initial uptake of SP. Other factors such as interpersonal, individual and socio-cultural factors that influenced uptake of IPTp-SP were explored.

Organisational arrangement such as assigning a health provider to provide SP and face-to-face interaction between health providers and ANC clients, created an enabling environment for women to take SP. These arrangements facilitated uptake of the first dose of SP. Therefore, it is not surprising that IPTp-SP+1 uptake in Ghana has been increasing over the years [28, 29]. Similarly, Maheu-Giroux and Castro [46] found that facilities that reported that IPTp-SP was routinely offered as part of ANC services were more successful in getting women to receive optimum doses of SP. Also, Onyango-Ouma, Okuonzi [47] in a study in Kenya, found that the most sustainable and effective way to implement IPTp-SP was by integrating it into ANC service delivery.

All hospitals and healthcare centres enforced DOT, which was effective, as it contributed to adherence and a high uptake of IPTp-SP1. Similarly, other studies reported that women who attended health facilities, where DOT was practiced, were more likely to receive at least one dose of IPTp-SP [48–50]. Other studies have found that poor supervision of SP intake by health providers contributed to low adherence and uptake of IPTp-SP among pregnant women [9, 18]. Another study also found that delivering IPTp through ANC was ineffective especially in higher level facilities. The two intermediate steps that were noted to be ineffective in the delivery of IPTp at that level were women attending ANC being given any SP and being given IPTp by DOT [51].

Another health system factor that promoted IPTp-SP uptake was recording the monthly uptake of SP in the client's maternity booklet. This ensured that healthcare providers could quickly refer to women's history on IPTp-SP such as women who were eligible for IPTp-SP and those who were due to take SP during an ANC visit, which facilitated quick clinical decision to offer such women IPTp-SP. Even women who traveled out of their original place of residence were able to continue with their monthly uptake of SP seamlessly in other health facilities. They only needed to present their maternity booklets in their second facilities and the healthcare provider referred to the booklets to know their history to facilitate clinical decision making. A study in South Africa equally found that a new individual maternity case record card introduced in maternal healthcare delivery greatly improved the referral process and contributed to quality health service delivery, however it increased the workload of midwives [52].

Study participants who were unwilling to take SP for fear of side-effects were influenced to take it after healthcare providers convinced them by providing them with information. Women in one facility visited herbalists or took herbs, while several women in all the study communities visited prayer camps and followed pastors' directives. Such attitudes contributed to women skipping scheduled ANC visits or starting ANC late, which contributed to low uptake of SP. Similarly, a study noted that when women are counseled and given information on malaria and IPTp-SP it increases uptake of SP [20]. Other studies have also reported on alternative sources of care that women use, which hampers their ability to utilize maternal health care services optimally [53, 54].

Health providers appeared to have devised simple but effective ways of dealing with individual, socio-cultural and interpersonal challenges that pregnant women navigate through in their daily interactions. Strategies that facilitated SP uptake included insisting that women ate before taking SP and that women brought water, as well as informing clients to carry along water to take SP in subsequent ANC visits. Another strategy was encouraging women to negotiate with family members to postpone or forgo responsibilities that could cause them to miss scheduled ANC visits. Health workers also discouraged women from obeying pastors who directed pregnant women to fast. Health workers' approach helped to dispel misconceptions, superstitious beliefs about pregnancy and cultural practices, which discouraged women from accessing SP. This ensured that most women who attended ANC took SP whenever available. Similarly, previous studies found that perceptions that SP should not be taken without eating

and lack of clean water and cups in ANC clinics hindered the effective implementation of DOT in health facilities [9, 55, 56]. A study in Southern Tanzania found that women who had previously taken SP on empty stomachs and reacted badly such as vomiting and becoming weak, refused to take it, unless it was given under DOT [57]. Hill, Hoyt [11] and Webster, Kayentao [58] reported that side effects of SP contributed to ineffective delivery in Mali. Studies in Tanzania also reported that some pregnant women avoided and postponed ANC attendance because of the fear of side-effects of SP on the foetus [59]. Nevertheless, a study in Uganda reported the contrary [60]. The study found that most women considered SP to be safe and were willing to take it again in future without supervision, despite experiencing unwanted effects of SP on previous occasions [60]. Doku, Zankawah's [56] study found that ANC staff had limited knowledge on the dropout rate in their catchment area and poor attitudes of some health workers contributed to low uptake of IPTp-SP3.

This study found that stock-outs of SP in health facilities affected regular uptake of IPTp-SP in some facilities. This is worrying, because most ANC providers were willing and effective in administering SP under DOT. Aberese-Ako, Magnussen (38) in their earlier study found that some facilities dealt with stock-outs by compelling women to buy SP from private pharmacies, others sold SP to clients, while one facility rationed SP to clients. Such decisions were borne out of health facility managers' decisions to address stock-outs by charging fees and rationing SP, which was reported in the earlier paper [38]. Aberese-Ako, Magnussen [38], also reported that other facilities were reluctant to buy SP from the open market, because of a directive from the Ghana Health Service, which mandated them to buy drugs from the CMS. They could buy drugs from the open market only when the CMS did not have the required drug. Other facilities did not have money to buy SP from the open market, so they depended heavily on the CMS. The decision to charge fees contributed to women who could not afford to pay for ANC services skipping scheduled visits and others skipping some of the ANC procedures. Also, some of the study participants who attended ANC regularly did not complete the recommended 5–7 doses of SP due to stock-outs of SP at the Central Medical Stores (CMS) and the rationing of SP [38]. Similarly, other studies have reported that shortages of SP in ANC clinics reduced uptake of SP [9, 46, 53, 56]. Another study has attributed low uptake of SP to lack of services and lack of sufficient staff to implement the IPTp-SP policy [61]. Similarly, studies in other parts of sub-Saharan Africa reported that high cost of ANC services and MiP interventions including SP demotivated women from accessing ANC, which affected access to SP [57, 62, 63].

Some women in this current study could not afford the cost of ANC services such as paying for laboratory test, buying the maternity booklet and buying SP. Such women sometimes skipped some of the ANC procedures especially laboratory tests, delayed initiating ANC and skipped their ANC scheduled appointments, which contributed to low uptake of SP. Similarly, studies conducted in Uganda and Mali found that cost of accessing ANC and SP at health facilities in both public and private facilities, hampered utilization of ANC services including IPTp-SP uptake [64–66]. Also, studies have found that early and frequent ANC visits increased uptake of IPT-SP3+ among married women, while early ANC attendance contributed to IPTp-SP3+ uptake among rural women [67, 68]. Another study in Ghana noted that women who began ANC in the third trimester of pregnancy, were not likely to meet the 3+ doses required, while those who initiated ANC in the first and second trimesters were more likely to meet the required 3+ doses of SP at the time of delivery [32].

While health workers indicated in interviews that they gave ANC clients ample information on SP, observations in ANC consulting rooms and IDIs with women revealed that one in two ANC clients did not receive information, so they did not know when they were supposed to start taking SP, why they were taking SP and the optimum doses recommended for the

duration of a pregnancy. Nonetheless, some healthcare providers admitted that one in two women were not given ample information, because of heavy workload. Also, some health workers were unclear about the recommended optimum doses of IPTp-SP and how long women were allowed to take it, so it is not surprising that women did not also know. This gap in information affected early uptake and regular intake of SP, as women were not motivated to return to facilities after the first dose. It also explains why many women lacked information on the change in policy from 3 to 5 or more doses of SP, which may have accounted for the default. Another factor could be the volume of clients who attended ANC. Most facilities had two to three midwives attending to an average of 60 women in a day. This was quite tasking and time consuming, because the facilities practiced focused ANC, which required that each patient was given integrated services and should have ample interaction with the healthcare provider. The healthcare provider was also required to document some of the processes such as recording IPTp-SP uptake among others. Such challenges are not uncommon in lower and middle income countries, where health facilities and patient-health provider ratio tend to be very high [69–71]. Aburayya, Alshurideh's [70] study in Dubai and Bradley, Kamwendo's [71] study in Malawi found that few health care facilities coupled with too few staff and too many patients contributed to poor quality service delivery and health worker burnout. In advanced countries like the United Kingdom it has also been found that communities that had a right balance of healthcare facilities, healthcare providers and fewer patients experienced lower levels of mortality compared to communities that had fewer facilities and healthcare providers but high patient load [69]. Other studies in Ghana and Tanzania found that a good number of pregnant women did not know why they were given SP [5, 72]. Some studies reported that pregnant women who were attending private and public clinics lacked correct knowledge on the effect of malaria in pregnancy and the benefits of SP in pregnancy [60, 73]. Consequently, studies have concluded that women's lack of understanding and correct knowledge of IPTp-SP contributed to low uptake and adherence [53, 54, 60, 74].

Refusing to access IPTp-SP without any justification affected uptake. Interestingly, most of those who refused to take SP were teachers who are well-educated. This is probably because being educated, they were aware of their right to refuse and for that matter the health workers could not compel them to take SP. Also, the educated appeared to rely on information sources such as information from the internet that discouraged them from taking SP. Such an attitude contributed to low uptake of SP despite high ANC attendance in Ghana [29, 30]. Similarly, a study reported that well educated ANC clients such as teachers expressed fear of SP and were reluctant to take SP. They were equally influenced by media reports on side effects of SP [59]. Rassi, Graham [74], reported that reluctance to take in medication served as a barrier to IPTp-SP uptake. Marchant, Nathan [75], on the other hand, found no evidence of individual factors affecting second dose coverage beyond living in an urban area. Exavery, Mbaruku [20] and Okethwangu, Opigo [76], reported the contrary that women with secondary or higher education compared to the less educated were more likely to take optimal doses of SP during pregnancy.

High trust in health facilities and health care providers influenced initial and regular uptake of IPTp-SP. Even women who lacked knowledge on the reasons why they were taking SP, were motivated to take it, due to their firm belief that health providers meant well. ANC clients' trust in health providers and the healthcare system is crucial for building an effective health provider-client relationship [77]. Such a relationship facilitates clients' adherence to treatment, satisfaction with care received and clients returning to health facilities for maternal health care services including completing the recommended 5 or more doses of SP [38, 77–79]. Similarly, studies in Nigeria, Southern Mozambique and Uganda found that women who trusted health providers were highly motivated, so they willingly took in drugs such as SP, which were

prescribed by health providers [49, 53, 74, 80]. Trust has been noted as crucial in the emotional and interpersonal aspects of the patient-health provider relationship and ensures effective therapeutic encounters. It affects a host of important patient behaviours and attitudes relating to care including seeking care, patient disclosure of critical medical information, complying with treatment and patient satisfaction with care [81]. Yevoo, Agyepong [82], reported that ANC clients' lack of trust in health providers resulted in clients lying to healthcare providers about their reproductive history, which affected clinical decisions. Studies on healthcare provision in South Africa have also reported on the critical role trust plays in client-healthcare provider encounters and its influence on adherence to treatment [77, 83]. In Uganda on the other hand, it was found that a change in healthcare policies resulting in a drop in government subvention to healthcare facilities contributed to the introduction of user fees for service. This development resulted in distrust of healthcare facilities and a rise in self-medication [84].

Women whose husbands, other family members and friends supported them with funds or encouraged them to attend ANC and to take IPTp-SP completed 5 doses of IPTp-SP. Those who were discouraged from attending ANC did not complete the optimum doses of IPTp-SP. This suggests that interpersonal relations influence women's health seeking behaviour and thus should be considered in future interventions. Other studies corroborate this finding [53, 66, 85]. The different studies noted husbands' encouragement, financial support and reminder to wives to take their drugs influenced uptake of IPTp-SP and adherence to other maternal healthcare demands. Also, extended family members and neighbors influenced pregnant women's utilization of ANC and SP services [53, 66, 85]. The influence of interpersonal relations in facilitating acceptance of healthcare interventions has also been reported in interventions on long lasting insecticide treated bed nets [86, 87]. Other studies in sub-Saharan Africa attest to the crucial role of male involvement in HIV treatment and prevention [88], maternal and child health services [89], as well as in family planning decision making [90].

Only one facility reported that community health workers were encouraged to provide SP to women who were continuing IPTp-SP in communities. This approach ensured that women who experienced challenges with distance, ANC charges and family commitments could still access IPTp-SP. Similarly, other studies have noted the importance of using community health workers in the administration of IPTp-SP in Africa [91, 92]. Mbonye, Hansen [92], found that using community health workers to deliver IPTp-SP and other maternal health services in communities was effective. In Ghana, community health officers are trained health professionals, who carry out outreach programmes in communities. This gives them access to communities and for that matter they are in a good position to reach out to women who fail to honor ANC visits. The Volta regional health directorate in its efforts to promote uptake of IPTp-SP has recently rolled out an intervention that seeks to use the CHPS programme to deliver IPTp-SP service in communities. This involves midwives working with community health officers (CHOs) to improve access to IPTp-SP service through ANC outreaches, home visits, defaulter tracing and follow-ups [93].

Unmarried young women and adolescents initiated IPTp-SP late and thus could not complete 5 or more doses. Most of them were brought to the ANC by their mothers and grandmothers, which suggests that they were probably forced to attend ANC and they lacked funds to access ANC independently. Besides, most of them did not receive support from the men responsible for the pregnancies. This is worrying, because most of them were experiencing pregnancy for the first time, yet they were likely not to have access to nutritious meals and were more likely to suffer from negative consequences of malaria in pregnancy such as abortions and still births [94, 95]. Facility ASF02's strategy of dedicating one day in a week to offering ANC services to adolescents and giving them a drink as an incentive is commendable. There is ample evidence to suggest that adolescents are reluctant in accessing key

interventions, which is not surprising [96–98]. Factors such as lack of money, shame in sharing space with their parents, cultural norms that place them in subordinate positions, socio-cultural taboos among others affect their willingness to access even interventions like sexual and reproductive health services [96–98]. Similarly, other studies revealed that adolescents were the least likely to complete at least three ANC visits, resultantly they were less likely to adhere to IPTp-SP uptake [61, 99, 100]. This is a result of structural constraints such as their low position in society, most of them being unmarried, sometimes they feel embarrassed to use ANC services with older women and they also fear that others will gossip about them [11, 61, 99, 100]. The contrary was found in a study in Tanzania, which reported that despite most adolescents being single, they attended ANC with older women [61]. The study found that socio-cultural factors such as late recognition of pregnancy and not being supported by the husband or partner influenced late antenatal care enrolment [61]. Another study reported the contrary that the proportion of women making more ANC visits decreased with increasing parity [54].

This study is limited in its ability to generalize the findings, as data was collected in only eight health facilities. However, the findings compare with previous studies that have been carried out in other context. Additionally, the study is subject to researchers biases in using the observation data collection method, because researchers could misinterpret interactions between healthcare providers and patients. In order to minimize this bias multiple data collection methods were used, which were later triangulated. Also, revisits were conducted to selected study participants to seek clarification over incidents that were difficult for the researchers to interpret.

## Conclusion

This study found that multiple factors such as how healthcare is organized and health providers approach to administering IPTp-SP, individual and socio-cultural factors influenced IPTp-SP uptake in two Ghanaian regions. Consequently, interventions that seek to address IPTp-SP uptake need to involve the healthcare system, ANC clients and the community.

The findings suggest that while integrating IPTp-SP into the health care delivery system influences initial uptake of IPTp-SP, it does not necessarily ensure that women take optimum doses of IPTp-SP. Health providers' approach to dealing with individual, interpersonal and socio-cultural barriers to IPTp-SP uptake is commendable and should be replicated in other settings. This will help to address individual and socio-cultural challenges in administering IPTp-SP. However, health providers' failure to provide clients with adequate information on SP is worrying. Providing women with the right information before or after the first dose of SP is one of the key factors that motivate women to return for subsequent doses. Consequently, health facilities need to encourage health providers to provide ANC clients with relevant information on malaria and IPTp-SP. Some women's willingness to buy SP suggest that if women are well informed of the relevance of SP, some would be motivated to prioritize buying and taking IPTp-SP unsupervised.

Health facilities and health workers can only be effective in facilitating IPTp-SP uptake, if facilities are regularly provided with SP at no cost. Thus, systemic challenges such as stock-outs need to be addressed. The Ghanaian government through the MoH should resume the supply of maternity record booklets and other medical essentials to both public and faith-based facilities for onward distribution to women free of charge. This would help to defray the cost of ANC services and thus encourage women to partonise them. In addition to this initiative, the effective implementation of the fee-free maternal health delivery service will promote early and regular attendance to the ANC, which will go a long way to ensure that women receive optimum doses of IPTp-SP.

Health facilities would also be able to facilitate IPTp-SP uptake by involving community health workers and volunteers to carry out follow-up visits in communities. The decision of the Volta Regional Health Directorate to extend IPTp services through the CHPS programme is commendable and it is hoped that other regions in Ghana would one day adopt the intervention. Additionally, health providers and the state with the support of the media need to encourage early initiation of ANC among pregnant women. Early initiation of ANC could contribute to women being able to complete 5 or more doses of IPTp-SP. It will help to enforce uptake of maximum doses of IPTp-SP, because women who are not able to honor scheduled ANC visits will be able to receive SP in communities.

Adolescents are a critical group that need ample attention and care. Both public and faith-based facilities need to dedicate one ANC day each week to adolescents, as one of the facilities in the Ashanti region has done, which is commendable. The study team observed more adolescents patronizing ANC service in the facility than in the other 7. This suggest that if a day is dedicated to them they will be motivated to utilize ANC services, since the fear of stigmatization, shame will be less and they will also enjoy some privacy. Additionally, maternity fees need to be waived for them, since most of them are unemployed and cannot afford to pay, which is one of the deterrents to their utilization of ANC services. Such interventions will facilitate early and consistent ANC attendance, which will facilitate IPTp-SP uptake among young women and adolescents.

## Supporting information

**S1 File. SL 1 observation checklist used in the study.**
(DOCX)

**S2 File. SL 2 pregnant women's interview guide.**
(DOCX)

**S3 File. SL 3 health provider's interview guide.**
(DOCX)

**S4 File. SL 4 health manager's interview guide.**
(DOCX)

## Acknowledgments

We wish to thank the nine research assistants, ED who supported the coding process, the regional health directorates and regional directors of health, participating districts, facilities, communities and study participants, for the cooperation and support in this study. We also thank the editor and the two reviewers whose inputs have contributed greatly to improve the quality of the manuscript.

## Author Contributions

**Conceptualization:** Matilda Aberese-Ako, Pascal Magnussen, Gifty D. Ampofo, Harry Tagbor.

**Formal analysis:** Matilda Aberese-Ako.

**Investigation:** Matilda Aberese-Ako.

**Methodology:** Matilda Aberese-Ako, Margaret Gyapong, Evelyn Ansah.

**Resources:** Matilda Aberese-Ako.

**Software:** Matilda Aberese-Ako.

**Supervision:** Matilda Aberese-Ako, Pascal Magnussen, Margaret Gyapong, Evelyn Ansah, Harry Tagbor.

**Writing – original draft:** Matilda Aberese-Ako, Margaret Gyapong.

**Writing – review & editing:** Matilda Aberese-Ako, Pascal Magnussen, Gifty D. Ampofo, Margaret Gyapong, Evelyn Ansah, Harry Tagbor.

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
