## [Decision Letter · Decision Letter 0]

26 Apr 2021

PONE-D-20-36566

An ethnographic study of how health system, socio-cultural and individual factors influence uptake of intermittent preventive treatment of malaria in pregnancy with sulfadoxine-pyrimethamine in a Ghanaian context

PLOS ONE

Dear Dr. Aberese-Ako,

Thank you for submitting your manuscript to PLOS ONE. After careful consideration, we feel that it has merit but does not fully meet PLOS ONE’s publication criteria as it currently stands. Therefore, we invite you to submit a revised version of the manuscript that addresses the points raised during the review process.

We look forward to receiving your revised manuscript.

Kind regards,

R Matthew Chico, MPH, PhD

Academic Editor

PLOS ONE

Journal Requirements:

3. Please state in your methods section whether you obtained consent from parents or guardians of the minors included in the study or whether the research ethics committee or IRB approved the lack of parent or guardian consent.

4. Please ensure that your recent study (Aberese-Ako, Matilda, et al. "Managing intermittent preventive treatment of malaria in pregnancy challenges: an ethnographic study of two Ghanaian administrative regions." Malaria journal 19.1 (2020): 1-17.) is mentioned  in the Introduction and in the Discussion, and that you have adequately discussed the relationship between the two studies.

5. Please provide additional details regarding participant consent. In the ethics statement in the Methods and online submission information, please ensure that you have specified how verbal consent was documented and witnessed.

6. When reporting the results of qualitative research, we suggest consulting the COREQ guidelines: http://intqhc.oxfordjournals.org/content/19/6/349. In this case, please consider including more information on the number of interviewers, their training and characteristics. Moreover, please provide the interview guide used as a Supplementary file.

7. We note that you have indicated that data from this study are available upon request. PLOS only allows data to be available upon request if there are legal or ethical restrictions on sharing data publicly. For information on unacceptable data access restrictions, please see http://journals.plos.org/plosone/s/data-availability#loc-unacceptable-data-access-restrictions.

8. Please include captions for your Supporting Information files at the end of your manuscript, and update any in-text citations to match accordingly. Please see our Supporting Information guidelines for more information: http://journals.plos.org/plosone/s/supporting-information

9. Your ethics statement should only appear in the Methods section of your manuscript. If your ethics statement is written in any section besides the Methods, please delete it from any other section.

Reviewers' comments:

Reviewer's Responses to Questions

**Comments to the Author**

1. Is the manuscript technically sound, and do the data support the conclusions?

Reviewer #1: Yes

Reviewer #2: Yes

2. Has the statistical analysis been performed appropriately and rigorously? 

Reviewer #1: N/A

Reviewer #2: N/A

3. Have the authors made all data underlying the findings in their manuscript fully available?

Reviewer #1: Yes

Reviewer #2: Yes

4. Is the manuscript presented in an intelligible fashion and written in standard English?

Reviewer #1: Yes

Reviewer #2: Yes

5. Review Comments to the Author

Reviewer #1: This is a very interesting and well written study, drawing upon a rich data source to explore reasons behind uptake or lack of uptake of IPTp-SP in Ghana. A few specific comments are provided below:

Background

• Lines 82-83: It seems unfair to say the findings were inconclusive because they were based on reported data – we rely on this type of data for many studies. It does of course have limitations in comparison to observations – suggest rephrasing along those lines and use this to emphasise the importance and value added of your study.

Methods

• Lines 127-128: perhaps add something about the addition of this HF leading to the total of 5 districts (rather than 2+2 as suggested by stage I of the four-stage selection process.

• It would be useful to describe how IDIs differ from conversations. Were conversations recorded and transcribed? Or did the RAs take notes that were used for the analysis?

• Are the case studies in table 2 the ANC attendants randomly selected and accompanied through the ANC process (lines 189-190)? If so, would be useful to state that in the main text.

Results

• How did you identify the number of doses of IPTp-SP that women received? Did you stratify the analysis by women with/without 5 doses?

• As this was a longitudinal study, did you observe/interview the same women or other participants more than once?

• Line 273: how did the HWs keep a record of who was due to have IPTp-SP on a given day (you may mention it elsewhere)? This seems useful information for achieving high coverage.

• Line 442: advise not advice

Discussion

• Lines 574-576: this sentence sounds out of place in the first paragraph (especially as overall this does not seem to be one of the more important influences on lower uptake). I suggest you remove it from here as you discuss it further down in the discussion.

• Line 589:”Others found the contrary was true” is a bit misleading – I think you mean that other studies found that lack of supervision resulted in low uptake (rather than other studies finding that DOT resulted in low uptake). Both support the same point that the current successful focus on DOT should be continued.

• Line 591-592: The finding form study #41 needs expanding slightly – what were the factors that meant delivery of IPTp through ANC in this setting was ineffective?

• Line 593: “The finding suggests…” Which finding are you referring to? Something from your study?

• Overall, your results section suggests that health system factors were generally positive in terms of uptake or IPT-SP, with the exception of stock outs of SP. In fact, even the lack of information did not seem to be a barrier to most women (the exception perhaps being teachers, or other more highly educated women?) – if they were told to take the SP then they took it. Your discussion of this and the role that health workers play in providing an enabling environment is well made. Trust of health providers is crucial as you discuss and it seems that your study area is leading the way on these measures compared to other studies on IPTp uptake. I note that you commend this in the conclusion which is good to see.

• However, as you say – attitudes that led to late attendance had a big impact on achieving higher number of doses. Is it information about malaria and IPTp that will lead to early attendance? Or perhaps other messaging or interventions (such as having adolescent specific days, encouragement from husbands/family members/community leaders) could effectively encourage early attendance and the uptake of IPTp would follow (health system factors permitting)

• Line 639: duplication of “which contributed”

• Lines 662-663: what were the information sources discouraging IPTp? Was it social media? Could expand on this slightly as these channels could potentially be used to deliver positive messages.

• Lines 671-676: these sentences on late attendance seem out of place with the rest of this paragraph which is discussion the potential link between higher education levels and refusal to take IPTp-SP.

Conclusion

• You make a clear summary of your findings and there are some interesting and practical recommendations. However, mention of encouraging early attendance is missing. Encouraging early attendance is critical to ensuring the full number of IPTp-SP doses are taken – I suggest including this in your conclusion, along with potential ways to do this.

Reviewer #2: I read with great interest this study entitled “An ethnographic study of how health system, socio-cultural and individual factors influence uptake of intermittent preventive treatment of malaria in pregnancy with sulfadoxine-pyrimethamine in a Ghanaian context”. Acknowledging the well-known gap between ANC attendance and IPTp coverage, this paper assesses the multiple factors influencing IPTp-SP uptake in 2 regions in Ghana. Based on ethnography, the manuscript reports relevant information on the social component of antenatal care that, ultimately, could be of great interest when designing interventions aiming at improving access to maternal health services.

Overall, the study identifies an important topic to address in public health interventions and, indeed, this is a nice piece of example showing how anthropological approaches can contribute to the body of knowledge related to malaria in pregnancy.

However, there are some aspects I would like the authors to consider in order to improve manuscript before publication.

General comments:

The manuscript is well-written, well performed in terms of methodology, easy to read and clear results and discussion sections. The topic of the study is of highly relevance and, while the study appears to be sound, I would suggest the authors to stress the rational of the study in the Background section, given the quite extensive literature existing on determinants of IPTp uptake or ANC attendance.

Moreover, the objective of the study, as it is stated in the text (lines 106-108) seems to directly refer to the identified factors, which seems a bit confusing: If the aim of the study was to explore “how health system, individual and socio-cultural factors influenced IPTp-SP uptake”, were, then, these categories previously identified? I would recommend the authors to either rephrase or to clarify the objective.

I would like to remark the efforts of the authors to describe the methodological procedures and the decisions made by the research team, including community entry activities. I appreciate the inclusion of the observation checklist and interview guides as additional material. However, I advise the authors to clearly describe how the urban/rural criteria to select the settings were defined, in order to clarify what contextual characteristics were taken into consideration. Finally, I would only recommend to rephrase the Data collection techniques subsection, so that the reader could easily understand what techniques were used and what group of people they were targeting (and information that it is already presented in table 2).

I would suggest the authors to structure the results section, by adding subheadings, following the 3 categories of analysis. It would be helpful for the reader to understand how the researchers relate the subcategories to the broader categories of analysis (health system, individual and socio-cultural factors). For instance, the subsection called “Women knowledge…” is not clear whether it is part of the health system or individual factor (which, to me, would have made more sense, but it seems a category that appears later in the section).

I would like to emphasize the particular value of providing pieces from observation notes, together with the quotations. In this respect I would require the authors to revise the identification of the quotations presented throughout the text, since there is missing information in some cases (e.g. type of technique or the respondent profile). It would also be helpful to specify how the health facility pseudonym will be presented (providing an example in line 226 could be an option).

In the discussion section, I have the impression that authors present some findings that are not included in the results, for example what is presented in page 31 (lines 593-595, about the type of information provided to pregnant women). I would ask the authors to revise this aspect.

I would have appreciated a more nuanced discussion of the results. I think that the discussion section does not only needs to present how previous works support (or not) the results but, rather, it allows for going one-step beyond in the interpretation of data. Thus, I encourage the authors to expand the discussion section in order to provide a rich dialogue between the findings of the study, previous works and broader topics or other public health interventions. For instance, trust on health providers could be an element for further discussion, since it is an aspect which has deserved much attention in other studies related to other type of interventions (such us community engagement approaches). In the same vein, results related to interpersonal factors could also be discussed within the frame of gender roles, female decision-making processes and/or the role of community-shared meanings, values and norms.

Finally, including a paragraph on possible limitations of the study, could be an asset.

Specific comments:

Abstract

• Line 39: Revise the year and the WHO recommendation, which seems incorrect (see comment bellow)

Background:

• Lines 75-78: The year when the WHO recommendation is announced is missing in the text (but it is the abstract). However, I would ask the authors to revise it. As far as I know since 2012, WHO has recommended IPTp at each scheduled antenatal care clinic visit starting in the second trimester of gestation, with the objective of ensuring the uptake of at least three IPTp administrations of SP. The references provided to support that the recommendation is to provide a minimum of 5 doses are not from WHO but, rather, other papers. I would ask to revise if this is a National Policy in Ghana, and change it accordingly.

Methods

• Line 115: I advise to elaborate the relation with the reference provided about the Hawthorne effect.

• Line 117: There’s a typo (informal conversions, which should be conversations)

• Table 2: I would spell out DHD officials, as a footnote on the table.

Results:

• Lines 293-294: there’s a repeated sentence (Health workers reported that education was critical for adherence)

• Lines 350-354: I’m not sure if these 2 quotes presented exemplify the previous statement.

• Lines 432-438: How the information provided in this paragraph correlates with the issue of SP stock-outs

• Line 453: “Interpersonal and socio-cultural factors also influenced uptake of IPTp-SP” is a sentence that already appears at the beginning of the subsection.

• Lines 482-484: I don’t understand how this relates to socio-cultural factors.

• Lines 539-544: I wonder to which extend this finding may not be related with sociocultural factors.

• Line 567: Specify from where this information comes from (and IDI? Conversation?).

Discussion

• Line 625: There a single square bracket. I don’t know if this is a typo or the authors have forgotten to include a reference.

• Lines 638-639: Does “Some women…” refer to the study participants?

• Lines 639-640: “which contributed” appears twice.

I hope that above comments would help the authors to improve the manuscript. I am so looking forward to read the article once published.

Kind Regards.

6. PLOS authors have the option to publish the peer review history of their article (what does this mean?). If published, this will include your full peer review and any attached files.

Reviewer #1: No

Reviewer #2: No

---

## [Author Response · Author response to Decision Letter 0]

8 Jul 2021

RESPONSE TO REVIEWERS

Response to Editor

Many thanks for your kind support. We are grateful. Please find below, a point by point response to your comments and that of the reviewers. 

Response to Editor’s comments

 Response: Article has been revised to meet the PLOS ONE's style requirements, including those for file naming.

 Response: All references have been reviewed and they are all correct and complete. No retracted paper has been cited. 

3. Please state in your methods section whether you obtained consent from parents or guardians of the minors included in the study or whether the research ethics committee or IRB approved the lack of parent or guardian consent.

 Response: Only one 16 year old was interviewed with the consent of her mother. Please see lines 

4. Please ensure that your recent study (Aberese-Ako, Matilda, et al. "Managing intermittent preventive treatment of malaria in pregnancy challenges: an ethnographic study of two Ghanaian administrative regions." Malaria journal 19.1 (2020): 1-17.) is mentioned in the Introduction and in the Discussion, and that you have adequately discussed the relationship between the two studies.

 Response: My recent study (Aberese-Ako, Matilda, et al. "Managing intermittent preventive treatment of malaria in pregnancy challenges: an ethnographic study of two Ghanaian administrative regions." Malaria journal 19.1 (2020): 1-17.) is mentioned in the Introduction and in the Discussion. Please see lines 124-136 and 843-852.

5. Please provide additional details regarding participant consent. In the ethics statement in the Methods and online submission information, please ensure that you have specified how verbal consent was documented and witnessed.

 Response: Additional details regarding participant consent indicated. Also, verbal consent process reported. Please see…. And footnote 3. 

6. When reporting the results of qualitative research, we suggest consulting the COREQ guidelines: http://intqhc.oxfordjournals.org/content/19/6/349. In this case, please consider including more information on the number of interviewers, their training and characteristics. Moreover, please provide the interview guide used as a Supplementary file.

 Response: COREQ consulted. The number of interviewers, their training and characteristics reported. Please see lines 151-154 and 157-160.

7. We note that you have indicated that data from this study are available upon request. PLOS only allows data to be available upon request if there are legal or ethical restrictions on sharing data publicly. For information on unacceptable data access restrictions, please see http://journals.plos.org/plosone/s/data-availability#loc-unacceptable-data-access-restrictions.

Response: There are no ethical or legal restrictions on sharing de-identified data. Contact information for ethics committee: Mr. Fidelis Anumu, REC administrator, email: rec@uhas.edu.gh

 Response: Data from this study is available upon reasonable request through the

 Response: Thank you very much. 

8. Please include captions for your Supporting Information files at the end of your manuscript, and update any in-text citations to match accordingly. Please see our Supporting Information guidelines for more information: http://journals.plos.org/plosone/s/supporting-information

Response: Supporting information document read and the appropriate revisions made. 

9. Your ethics statement should only appear in the Methods section of your manuscript. If your ethics statement is written in any section besides the Methods, please delete it from any other section.

Response: Ethics statement has been indicated only in the methods section. Please, see lines 272-296

RESPONSE TO REVIEWER #1

Many thanks for your expert comments and time, please find below a point by point response to your comments

Reviewer #1: This is a very interesting and well written study, drawing upon a rich data source to explore reasons behind uptake or lack of uptake of IPTp-SP in Ghana. A few specific comments are provided below:

Response: Many thanks for your commendation.

Background

• Lines 82-83: It seems unfair to say the findings were inconclusive because they were based on reported data – we rely on this type of data for many studies. It does of course have limitations in comparison to observations – suggest rephrasing along those lines and use this to emphasise the importance and value added of your study.

Response: We appologise unreservedly for that error. It has been corrected. Please see lines 87-91.

Methods

• Lines 127-128: perhaps add something about the addition of this HF leading to the total of 5 districts (rather than 2+2 as suggested by stage I of the four-stage selection process.

Response: Explanation for selecting five districts and eight health facilities indicated. Kindly see 125-129

• It would be useful to describe how IDIs differ from conversations. Were conversations recorded and transcribed? Or did the RAs take notes that were used for the analysis?

Response: Explanation of conversations and IDIs indicated. Please see footnotes 1 and 2

• Are the case studies in table 2 the ANC attendants randomly selected and accompanied through the ANC process (lines 189-190)? If so, would be useful to state that in the main text.

Response: Case studies explained. Please see page 196-201

Results

• How did you identify the number of doses of IPTp-SP that women received? Did you stratify the analysis by women with/without 5 doses?

Response: How number of doses of IPTp-SP that women received was determined explained. Please, see lines 286-293

• As this was a longitudinal study, did you observe/interview the same women or other participants more than once?

Response: Only case studies were observed for most periods of their pregnancy. Most of them the observation ceased after they gave birth. Please see lines 165 - 172

• Line 273: how did the HWs keep a record of who was due to have IPTp-SP on a given day (you may mention it elsewhere)? This seems useful information for achieving high coverage.

Response: How did the HWs keep a record of who was due to have IPTp-SP Please, see lines 334-351.

• Line 442: advise not advice

Response: Thank you. advice replaced with “advise”. Please see line 508

Discussion

• Lines 574-576: this sentence sounds out of place in the first paragraph (especially as overall this does not seem to be one of the more important influences on lower uptake). I suggest you remove it from here as you discuss it further down in the discussion.

Response: The sentences has been deleted. Please see lines 770-772

• Line 589:”Others found the contrary was true” is a bit misleading – I think you mean that other studies found that lack of supervision resulted in low uptake (rather than other studies finding that DOT resulted in low uptake). Both support the same point that the current successful focus on DOT should be continued.

Response: “Others found the contrary was true” Deleted and sentence revised. Please see line 786,

• Line 591-592: The finding form study #41 needs expanding slightly – what were the factors that meant delivery of IPTp through ANC in this setting was ineffective?

Response: Factors that contributed to delivery of IPTp through ANC in the referred study ineffective expanded. Please see lines 806-808.

• Line 593: “The finding suggests…” Which finding are you referring to? Something from your study?

Response: “The finding suggests” changed to “study participants”. Please see line 800

• Overall, your results section suggests that health system factors were generally positive in terms of uptake or IPT-SP, with the exception of stock outs of SP. In fact, even the lack of information did not seem to be a barrier to most women (the exception perhaps being teachers, or other more highly educated women?) – if they were told to take the SP then they took it. Your discussion of this and the role that health workers play in providing an enabling environment is well made. Trust of health providers is crucial as you discuss and it seems that your study area is leading the way on these measures compared to other studies on IPTp uptake. I note that you commend this in the conclusion which is good to see.

Response: Many thanks for the commendation. 

• However, as you say – attitudes that led to late attendance had a big impact on achieving higher number of doses. Is it information about malaria and IPTp that will lead to early attendance? Or perhaps other messaging or interventions (such as having adolescent specific days, encouragement from husbands/family members/community leaders) could effectively encourage early attendance and the uptake of IPTp would follow (health system factors permitting)

Response: Information on malaria and IPTp included. Please see line1008

Response: Information on the need to institute adolescent clinics included. Please see 1027-1034

• Line 639: duplication of “which contributed”

Response: Second “which contributed” deleted. Please see 807 & 808

• Lines 662-663: what were the information sources discouraging IPTp? Was it social media? Could expand on this slightly as these channels could potentially be used to deliver positive messages.

Response: Information sources discouraging IPTp included. Please see lines 897-898

• Lines 671-676: these sentences on late attendance seem out of place with the rest of this paragraph which is discussion the potential link between higher education levels and refusal to take IPTp-SP.

Response: Sentences on late attendance deleted. Please see lines 906-911

Conclusion

• You make a clear summary of your findings and there are some interesting and practical recommendations. However, mention of encouraging early attendance is missing. Encouraging early attendance is critical to ensuring the full number of IPTp-SP doses are taken – I suggest including this in your conclusion, along with potential ways to do this.

Response: Encouraging early attendance included. Please see lines 1035-1037.

RESPONSE TO REVIEWER #2

Many thanks for your expert comments and time, please find below a point by point response to your comments

Reviewer #2: I read with great interest this study entitled “An ethnographic study of how health system, socio-cultural and individual factors influence uptake of intermittent preventive treatment of malaria in pregnancy with sulfadoxine-pyrimethamine in a Ghanaian context”. Acknowledging the well-known gap between ANC attendance and IPTp coverage, this paper assesses the multiple factors influencing IPTp-SP uptake in 2 regions in Ghana. Based on ethnography, the manuscript reports relevant information on the social component of antenatal care that, ultimately, could be of great interest when designing interventions aiming at improving access to maternal health services.

Overall, the study identifies an important topic to address in public health interventions and, indeed, this is a nice piece of example showing how anthropological approaches can contribute to the body of knowledge related to malaria in pregnancy.

However, there are some aspects I would like the authors to consider in order to improve manuscript before publication.

General comments:

The manuscript is well-written, well performed in terms of methodology, easy to read and clear results and discussion sections. The topic of the study is of highly relevance and, while the study appears to be sound, I would suggest the authors to stress the rational of the study in the Background section, given the quite extensive literature existing on determinants of IPTp uptake or ANC attendance.

Response: Rationale of study indicated. Please see lines 131-146.

Moreover, the objective of the study, as it is stated in the text (lines 106-108) seems to directly refer to the identified factors, which seems a bit confusing: If the aim of the study was to explore “how health system, individual and socio-cultural factors influenced IPTp-SP uptake”, were, then, these categories previously identified? I would recommend the authors to either rephrase or to clarify the objective.

Response: The objective has been rephrased. Please see lines 128-130

I would like to remark the efforts of the authors to describe the methodological procedures and the decisions made by the research team, including community entry activities. I appreciate the inclusion of the observation checklist and interview guides as additional material. However, I advise the authors to clearly describe how the urban/rural criteria to select the settings were defined, in order to clarify what contextual characteristics were taken into consideration. 

Response: Criteria used for selection of districts rectified. Please, see page 164-177

Finally, I would only recommend to rephrase the Data collection techniques subsection, so that the reader could easily understand what techniques were used and what group of people they were targeting (and information that it is already presented in table 2).

Response: Data collection techniques subsection rephrased. Please see line 231

I would suggest the authors to structure the results section, by adding subheadings, following the 3 categories of analysis. It would be helpful for the reader to understand how the researchers relate the subcategories to the broader categories of analysis (health system, individual and socio-cultural factors). For instance, the subsection called “Women knowledge…” is not clear whether it is part of the health system or individual factor (which, to me, would have made more sense, but it seems a category that appears later in the section).

Response: results section restructured. Please see lines 288 – 725

I would like to emphasize the particular value of providing pieces from observation notes, together with the quotations. In this respect I would require the authors to revise the identification of the quotations presented throughout the text, since there is missing information in some cases (e.g. type of technique or the respondent profile).

Response: Type of technique or the respondent profile has been revised. Please see results section.

 It would also be helpful to specify how the health facility pseudonym will be presented (providing an example in line 226 could be an option).

Response: Health Facility pseudonym and community pseudonym used specified. Please see lines 284-287

In the discussion section, I have the impression that authors present some findings that are not included in the results, for example what is presented in page 31 (lines 593-595, about the type of information provided to pregnant women). I would ask the authors to revise this aspect.

Response: Lines 593-595, about the type of information provided to pregnant women revised. Please see line 820-823

I would have appreciated a more nuanced discussion of the results. I think that the discussion section does not only needs to present how previous works support (or not) the results but, rather, it allows for going one-step beyond in the interpretation of data. Thus, I encourage the authors to expand the discussion section in order to provide a rich dialogue between the findings of the study, previous works and broader topics or other public health interventions. For instance, trust on health providers could be an element for further discussion, since it is an aspect which has deserved much attention in other studies related to other type of interventions (such us community engagement approaches). In the same vein, results related to interpersonal factors could also be discussed within the frame of gender roles, female decision-making processes and/or the role of community-shared meanings, values and norms.

Response: A more nuanced discussion of the results has been included. Please see lines 938-949, 958-962, 984-988

Finally, including a paragraph on possible limitations of the study, could be an asset.

Response: Paragraph reporting on possible limitations included. Please see lines 1000-1006

Specific comments:

Abstract

• Line 39: Revise the year and the WHO recommendation, which seems incorrect (see comment bellow)

Response: Year in abstract revised. Please see line 39

Background:

• Lines 75-78: The year when the WHO recommendation is announced is missing in the text (but it is the abstract). However, I would ask the authors to revise it. As far as I know since 2012, WHO has recommended IPTp at each scheduled antenatal care clinic visit starting in the second trimester of gestation, with the objective of ensuring the uptake of at least three IPTp administrations of SP. The references provided to support that the recommendation is to provide a minimum of 5 doses are not from WHO but, rather, other papers. I would ask to revise if this is a National Policy in Ghana, and change it accordingly.

Response: Background information and date of WHO recommendation revised. Please see line 74. 

Ghana’s recommendation of 5 doses stated. Please see line 97

Methods

• Line 115: I advise to elaborate the relation with the reference provided about the Hawthorne effect.

Response: Hawthorne effect elaborated. Please see lines 154 & 155

• Line 117: There’s a typo (informal conversions, which should be conversations)

Response: Typo “conversions” corrected to “conversations”. Please see line 158.

• Table 2: I would spell out DHD officials, as a footnote on the table.

Response: DHD spelt out. Please see Table 2

Results:

• Lines 293-294: there’s a repeated sentence (Health workers reported that education was critical for adherence)

Response: Repeated sentence “Health workers reported that education was critical for adherence” deleted. Please see line 381. 

• Lines 350-354: I’m not sure if these 2 quotes presented exemplify the previous statement.

Response: Two quotes not exemplifying the previous statement deleted. Please see lines 452-456. 

• Lines 432-438: How the information provided in this paragraph correlates with the issue of SP stock-outs

Response: The information has been elaborated and given a new heading. Please see lines 400-414.

• Line 453: “Interpersonal and socio-cultural factors also influenced uptake of IPTp-SP” is a sentence that already appears at the beginning of the subsection.

Response: “Interpersonal and socio-cultural factors also influenced uptake of IPTp-SP” deleted. Please see line 573.

• Lines 482-484: I don’t understand how this relates to socio-cultural factors.

Response: The paragraph has been moved to individual factors. Please see lines 690-694

• Lines 539-544: I wonder to which extend this finding may not be related with sociocultural factors.

Response: The finding has been improved, restrucctured and given a new sub theme heading. Please see lines 614- 647. 

• Line 567: Specify from where this information comes from (and IDI? Conversation?).

Response: Source of information specified. Please see lines 779-780

• Line 625: There a single square bracket. I don’t know if this is a typo or the authors have forgotten to include a reference.

Response: Many thanks, typo corrected, see line 851

• Lines 638-639: Does “Some women…” refer to the study participants?

Response: “Some women” corrected to “some of the study participants”. Please see line 863

• Lines 639-640: “which contributed” appears twice.

Response: Second “which contributed” deleted. Please see 826

I hope that above comments would help the authors to improve the manuscript. I am so looking forward to read the article once published.

Kind Regards.

Response: Many thanks, we are grateful for your time and insights, which has helped to improve the manuscript

---

## [Decision Letter · Decision Letter 1]

7 Aug 2021

PONE-D-20-36566R1

An ethnographic study of how health system, socio-cultural and individual factors influence uptake of intermittent preventive treatment of malaria in pregnancy with sulfadoxine-pyrimethamine in a Ghanaian context

PLOS ONE

Dear Dr. Aberese-Ako,

Thank you for submitting your manuscript to PLOS ONE. After careful consideration, we feel that it has merit but does not fully meet PLOS ONE’s publication criteria as it currently stands. Therefore, we invite you to submit a revised version of the manuscript that addresses the points raised during the review process.

We look forward to receiving your revised manuscript.

Kind regards,

R Matthew Chico, MPH, PhD

Academic Editor

PLOS ONE

Journal Requirements:

Reviewers' comments:

Reviewer's Responses to Questions

**Comments to the Author**

1. If the authors have adequately addressed your comments raised in a previous round of review and you feel that this manuscript is now acceptable for publication, you may indicate that here to bypass the “Comments to the Author” section, enter your conflict of interest statement in the “Confidential to Editor” section, and submit your "Accept" recommendation.

Reviewer #1: All comments have been addressed

Reviewer #2: All comments have been addressed

2. Is the manuscript technically sound, and do the data support the conclusions?

Reviewer #1: Yes

Reviewer #2: Yes

3. Has the statistical analysis been performed appropriately and rigorously? 

Reviewer #1: N/A

Reviewer #2: N/A

4. Have the authors made all data underlying the findings in their manuscript fully available?

Reviewer #1: No

Reviewer #2: Yes

5. Is the manuscript presented in an intelligible fashion and written in standard English?

Reviewer #1: Yes

Reviewer #2: Yes

6. Review Comments to the Author

Reviewer #1: The authors have done a thorough job of addressing all reviewer comments and I recommend that this paper is accepted. My only minor point of note is to have one last check for typos in the new text (I noticed some in the track changes version but these may have been corrected in the clean version)

Reviewer #2: I read with great interest the revision of this manuscript.

Either the background and the rationale of the study are more contextualized now.

The inclusion of headings and subheadings make the results section clearer to the reader, and facilitates the understanding of the analytical categories identified and used by the researchers. I would only recommend to simplify subheadings, by using more synthetic concepts (e.g. the one used in lines 574-576 is too long).

The discussion section shows a deeper interpretation of the results, and includes more complex dialogues with previous studies, with makes the manuscript analytically richer.

Corrections (such the year of the WHO recommendation, and its adaptation in Ghana) and specifications on settings identifiers and data collection techniques have also been introduced.

7. PLOS authors have the option to publish the peer review history of their article (what does this mean?). If published, this will include your full peer review and any attached files.

Reviewer #1: No

Reviewer #2: No

---

## [Author Response · Author response to Decision Letter 1]

9 Aug 2021

RESPONSE TO REVIEWERS

Reviewer #1: 

Many thanks for your kind support. We are grateful. Please find below, a point by point response to your comment. 

Reviewer #1: The authors have done a thorough job of addressing all reviewer comments and I recommend that this paper is accepted. My only minor point of note is to have one last check for typos in the new text (I noticed some in the track changes version but these may have been corrected in the clean version)

Response: Thank you very much for your feedback and for excellent input. The entire manuscript has been read carefully and revised to correct all typos. 

Reviewer #2: 

Many thanks for your kind support. We are grateful. Please find below, a point by point response to your comment. 

Reviewer #2: I read with great interest the revision of this manuscript.

The inclusion of headings and subheadings make the results section clearer to the reader, and facilitates the understanding of the analytical categories identified and used by the researchers. I would only recommend to simplify subheadings, by using more synthetic concepts (e.g. the one used in lines 574-576 is too long).

Response: Thank you very much for the excellent comment. Several of the headings have been revised to simplify them by using synthetic concepts. Please see lines 307, 391, 436, 574-576, 580 and 674. 

The discussion section shows a deeper interpretation of the results, and includes more complex dialogues with previous studies, with makes the manuscript analytically richer.

Corrections (such the year of the WHO recommendation, and its adaptation in Ghana) and specifications on settings identifiers and data collection techniques have also been introduced.

Response: Many thanks, we are grateful for your time and insights, which has helped to improve the manuscript

---

## [Editor Report · Decision Letter 2]

8 Sep 2021

An ethnographic study of how health system, socio-cultural and individual factors influence uptake of intermittent preventive treatment of malaria in pregnancy with sulfadoxine-pyrimethamine in a Ghanaian context

PONE-D-20-36566R2

Dear Dr. Aberese-Ako,

We’re pleased to inform you that your manuscript has been judged scientifically suitable for publication and will be formally accepted for publication once it meets all outstanding technical requirements.

Kind regards,

Marianne Clemence, Staff Editor, PLOS ONE, on behalf of

Matthew Chico

Academic Editor

PLOS ONE
---

## [Editor Report · Acceptance letter]

24 Sep 2021

PONE-D-20-36566R2 

An ethnographic study of how health system, socio-cultural and individual factors influence uptake of intermittent preventive treatment of malaria in pregnancy with sulfadoxine-pyrimethamine in a Ghanaian context 

Dear Dr. Aberese-Ako:

I'm pleased to inform you that your manuscript has been deemed suitable for publication in PLOS ONE. Congratulations! Your manuscript is now with our production department. 

Kind regards, 

on behalf of

Dr. R Matthew Chico 

Academic Editor

PLOS ONE